# QArray: a GPU-accelerated constant capacitance model simulator for large quantum dot arrays

Barnaby van Straaten,[1, *] Joseph Hickie,[1] Lucas Schorling,[2]
Jonas Schuff,[1] Federico Fedele,[2] and Natalia Ares[2, †]

[1]*Department of Materials, University of Oxford,*
*Oxford OX1 3PH, United Kingdom*
[2]*Department of Engineering, University of Oxford,*
*Oxford OX1 3PH, United Kingdom*
(Dated: August 4, 2024)

## Abstract

Semiconductor quantum dot arrays are a leading architecture for the development of quantum technologies. Over the years, the constant capacitance model has served as a fundamental framework for simulating, understanding, and navigating the charge stability diagrams of small quantum dot arrays. However, while the size of the arrays keeps growing, solving the constant capacitance model becomes computationally prohibitive. This paper presents an open-source software package able to compute a $100 \times 100$ pixels charge stability diagram of a 16-dot array in less than a second. Smaller arrays can be simulated in milliseconds - faster than they could be measured experimentally, enabling the creation of diverse datasets for training machine learning models and the creation of digital twins that can interface with quantum dot devices in real time. Our software package implements its core functionalities in the systems programming language Rust and the high-performance numerical computing library JAX. The Rust implementation benefits from advanced optimisations and parallelisation, enabling the users to take full advantage of multi-core processors. The JAX implementation allows for GPU acceleration.

───────
\* barnaby.vanstraaten@kellogg.ox.ac.uk

† natalia.ares@eng.ox.ac.uk

## I. INTRODUCTION

Semiconductor quantum dot arrays are one of the leading platforms for the realisation of large-scale quantum circuits. As they grow in size and complexity, these arrays present exciting opportunities and significant challenges. Tuning and operating these devices involves finding precise gate voltages to achieve the desired charge occupancy and interdot coupling parameters. Charge stability diagrams provide a visual representation of the charge occupation in the quantum dot as a function of the applied gate voltages, allowing the identification and control of transitions between different charge configurations. The constant capacitance model is a widely-used equivalent circuit framework that models the electrostatic characteristics of quantum dot arrays, This model can be used to compute charge stability diagrams of quantum dots arrays[1–7]. As a result, the constant capacitance model has proven to be an invaluable resource for gaining insights into complex charge stability diagrams and training neural networks for automated tuning strategies [2, 5, 8–11].

However, the model encounters scaling limitations, making the simulations of arrays larger than four dots slow and impractical. Specifically, the problem of computing the lowest energy charge configuration in a brute force manner, as implemented in Refs. [2, 8], scales with $(n_{\max} + 1)^{n_{\mathrm{dot}}}$, where $n_{\max} \geq 1$ is the maximum number of charges considered at any of the quantum dots, and $n_{\mathrm{dot}}$ is the number of quantum dots being simulated.

We introduce `QArray`, an open-source software package for fast constant capacitance modelling, opening the path to the creation of ever-larger datasets for neural network training, as well as the realisation of digital twins, the use of machine-learning-informed tuning strategies, and the use of these simulators for tuning and characterisation in real-time. Our package contains two new algorithms to compute the ground state charge configuration, which we call `QArray` default and thresholded. These algorithms greatly reduce the number of calculations needed to find the ground state charge configuration by computing the energies of only the most likely rather than the full set of possible configurations. Our default algorithm's complexity scales with $2^{n_{\mathrm{dot}}}$. The thresholded algorithm sacrifices some accuracy to further reduce the number of charge states considered to $(1 + t)^{n_{\mathrm{dot}}}$, where $t \in [0, 1]$ is a threshold defining which charge states must be considered and which can be neglected. Both of our algorithms considerably reduce computational time, especially for larger arrays containing more charges. For example, our default algorithm, in the case of a five-dot array containing five charges, reduces the number of change states considered by at least 243 times compared to a brute-force approach.

Within `QArray`, the brute-force and our default algorithm are implemented in Python, Rust and JAX, (see Fig. 1b). The threshold algorithm is implemented in Python and Rust. The Rust implementations are optimised for parallel computation on a CPU, using `rayon` [12–14]. At the same time, a GPU can accelerate the JAX implementations through just-in-time (jit) XLA compilation [15]. The Python implementations offer no practical advantages over those written in Rust or JAX but are maintained for benchmarking and comparison testing. The Rust and JAX implementations of our algorithms can model arrays with 16 or more dots in seconds and simulate the charge stability diagrams of smaller arrays in milliseconds.

The following section details the matrix formulation of the constant capacitance model. In Section III, we discuss our algorithms and how they compute the charge ground state in both open and closed quantum dot arrays. Section IV covers how realistic features can be incorporated into the simulation and some useful analytical results. Section V provides a

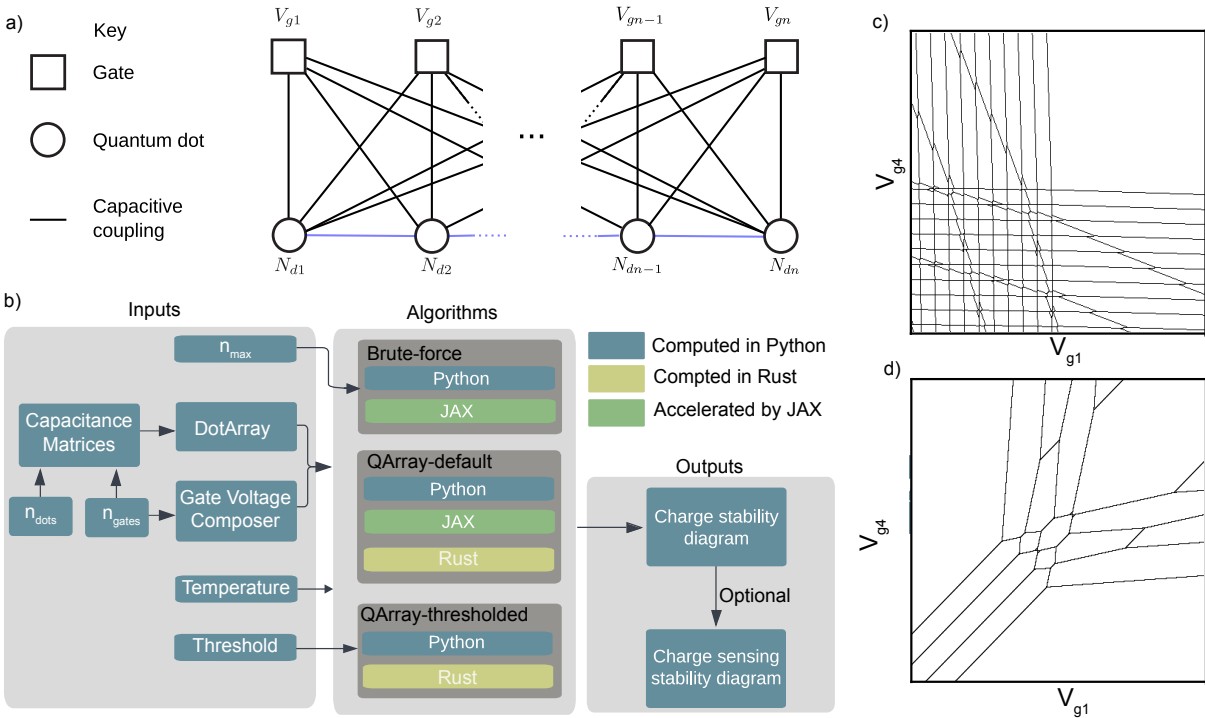

Figure 1. (a) Schematic of a generic array of quantum dots: The diagram illustrates the representation of a quantum dot device as a system of capacitively coupled electrostatic gates and quantum dots. Solid black lines represent the direct dot-to-gates couplings, while capacitors representing the interdot capacitance are modelled with blue lines, respectively. From the choice of the capacitance matrix (stored in the DotArray class), the user can define an array with an arbitrary number of dots and geometry. For given gate voltages, a discrete integer number of charges minimises the system's free energy. (b) The structure of the QArray package featuring the multiple software implementations, represented as colour-coded boxes denoting Python (blue), Rust (yellow), and JAX (green), of the brute-force, default and thresholded algorithms. The gate voltage composer class generates voltage vectors for native sweeps (e.g., $V_{g1}$ against $V_{g2}$), or it can be used to define arbitrary virtual gate sweeps. DotArray offers methods for computing charge stability diagrams for open or closed regimes with any core choice. Optionally, the charge stability diagram can be coupled with simulated charge sensors to model their response. (c) Charge stability diagrams (computed with our default algorithm implemented in Rust) for a hole-based linear array of four quantum dots in the open regime. The black lines denote transitions separating regions in the charge stability diagram with a fixed dot occupation. (d) Charge stability diagrams (computed with our default algorithm implemented in Rust) for the same linear four-dot array of quantum dots in the closed regime where the total number of charges in the array is fixed to four.

step-by-step example of using our software package to simulate a double quantum dot and demonstrates that `QArray` can be used to recreate experimentally-measured charge stability diagrams of open quadruple dot and closed five dot arrays. In Section VI, we discuss the details of our software implementations and how we optimised for speed. Finally, in Section VII, we benchmark our implementations of each algorithm.

## II. THE CONSTANT CAPACITANCE MODEL

The constant capacitance model describes an array of quantum dots and their associated electrostatic gates as nodes in a network of fixed capacitors [1–7], (see Fig. 1a). The model is based on the assumption that a single constant capacitance $C$ models the interactions between the charges on a single quantum dot and those in the environment and that the single-particle energy levels do not depend on the number of charges considered. Additionally, the model is a good approximation when the quantum dots are weakly coupled enough that quantum tunnelling effects can be ignored. Typically, the *constant* capacitance model is valid in a small range of gate voltages, as significant changes in the confinement potential can shift the dot positions and consequently alter their capacitive couplings. Each node $i$ has an associated charge $Q_i$, an electrostatic potential $V_j$, and its capacitive coupling to every other node $k$ given by $C_{jk}$. The capacitor $C_{ij}$ stores a charge $q_{ij} := C_{ij}(V_i - V_j)$. The total charge on node $i$ is therefore related to its and other nodes' potentials according to

$$Q_i = \sum_j q_{ij} = \sum_j C_{ij}(V_i - V_j). \tag{1}$$

Using the Maxwell matrix formulation, we express this as $\vec{Q} = \mathbf{c}\vec{V}$, where $\mathbf{c}$ is the Maxwell capacitance matrix. In this framework, the $\mathbf{c}$ matrix's diagonal elements give each node's total capacitance, and the off-diagonal elements are the negative of the capacitive coupling between the respective nodes. We now distinguish between quantum dots and electrostatic gate nodes by separating the matrix equation into

$$\begin{bmatrix} \vec{Q}_d \\ \vec{Q}_g \end{bmatrix} = \begin{bmatrix} \mathbf{c}_{dd} & \mathbf{c}_{gd} \\ \mathbf{c}_{dg} & \mathbf{c}_{gg} \end{bmatrix} \begin{bmatrix} \vec{V}_d \\ \vec{V}_g \end{bmatrix}, \tag{2}$$

where $\vec{Q}_{d(g)}$ represents the charge on the quantum dots (gates) and $\vec{V}_{d(g)}$ denotes the potential on the quantum dots (gates). The matrices $c_{dd}$, $c_{gd}$, $c_{dg}$, $c_{gg}$ encode dot-dot, gate-dot, dot-gate and gate-gate capacitive couplings, respectively, in the Maxwell format. The potential on the quantum dots can then be computed by

$$\vec{V}_d = \mathbf{c}_{dd}^{-1}(\vec{Q}_d - \mathbf{c}_{gd}\vec{V}_g). \tag{3}$$

The free energy of the system, $F(\vec{Q}; \vec{V}) = U(\vec{Q}; \vec{V}) - W(\vec{Q}; \vec{V}) = \vec{V}^T \mathbf{c}\vec{V}/2 - \vec{Q}_g\vec{V}_g$, can then be written as

$$F(\vec{Q}_d; \vec{V}_g) = \frac{1}{2}\vec{Q}_d^T \mathbf{c}_{dd}^{-1}\vec{Q}_d - (\mathbf{c}_{dd}^{-1}\mathbf{c}_{gd}\vec{V}_g)^T\vec{Q}_d, \tag{4}$$

where we have dropped the terms without dependence on $\vec{Q}_d$. This free energy function is convex since $\mathbf{c}_{dd}$ is positive definite (for proof, see Appendix B). Since the charge on the gates is not of interest, in the remainder of this paper, we drop the subscript and refer to the charge on the quantum dots as $\vec{Q}$.

The allowed charge states $\vec{Q} = \pm e\vec{N}$ (for holes and electrons, respectively) are heavily constrained in a quantum dot array. In particular, for an open quantum dot array, i.e. when the array is coupled to fermionic reservoirs from which it can draw an arbitrary number of charges (see Fig. 1 (c)), the constraints are:

(i) The number of charges must be integers, represented as $\vec{N} \in \mathbb{Z}^{n_{\text{dot}}}$.

(ii) The number of charges on any given quantum dot must be non-negative, meaning $N_i \geq 0$ for all $i$.

For a closed quantum dot array, i.e. when the leads are decoupled from the array, and the number of charges is fixed to a finite value $\hat{N}$ [16, 17] (see Fig. 1 (d)), we impose the additional constraint:

(iii) The sum of all charges in the array must equal $\hat{N}$.

At $T = 0$ K, the system will adopt the charge state, $\vec{N}^*$, which minimises the free energy whilst obeying the appropriate constraints, such that

$$\vec{N}^* = \underset{\vec{N} \in \mathcal{N}}{\arg \min} \, F(\vec{N}; \vec{V}_g), \tag{5}$$

where $\mathcal{N}$ is the set of all admissible charge configurations for the open or closed regimes of the quantum dot arrays. Due to the constraints, finding the lowest energy charge configuration is an example of a class of optimisation problems called constrained convex quadratic integer problems, also called Mixed-Integer Quadratic Programming (MIQP) problems, which are NP-hard [18–20].

By computing the ground state charge configuration as a function of the gate voltages, the constant capacitance model captures the charge stability diagrams of such quantum dot arrays when the tunnel couplings between quantum dots are weak. The following section discusses our algorithms for computing the charge configuration ground state.

## III.  THE QARRAY ALGORITHMS

The `QArray`-default and thresholded algorithms can be broken into two stages. First, we neglect the integer charge constraint, (i), and compute the continuously-valued charge state that minimises the free energy. We will refer to this state as the continuous minimum. In the second stage, we reintroduce the integer charge constraint by evaluating the energy of each discrete charge state that neighbours the continuous minimum and selecting the state with the lowest energy (Fig. 2 (a)).

The `QArray`-default and thresholded algorithms differ in which charge states they define as neighbouring the continuous minimum. The thresholded algorithm is much more selective (Sec. III B, Fig. 2 (b)). We formally justify considering only the nearest neighbouring discrete charge states in Appendix E. The fundamental idea is that the free energy is a convex function minimised at the continuous minimum. Therefore, the lowest-energy discrete charge states must lie close to the continuous minimum.

In the next section (Sec. III A), we will discuss how the continuous minimum is computed for both open and closed arrays, then how the default and thresholded algorithms find the discrete charge state lowest in energy from the nearest neighbours to the continuous minimum (Sec. III B).

### A.  Computing the continuous minimum

To compute the continuous minimum charge configuration for both open and closed quantum dot arrays, we initially try the analytical solution for the charge configuration

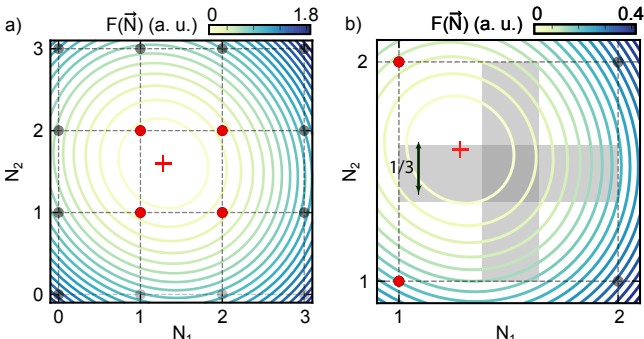

Figure 2. (a) Diagram depicting the default algorithm applied to an open double dot. The algorithm first finds the continuous minimum (marked with a red cross), which minimises the free energy. The free energy landscape, as a function of the charge on each dot, is shown with contour lines. After computing the continuous minimum, the algorithm evaluates the nearest neighbour discrete charge states (marked as red dots) to determine which is lowest in energy. By comparison, the brute force implementation considers all charge states, leading to it also having to consider the charge states marked as black dots. In this case, (1, 2) is the discrete charge state lowest in energy. (b) Diagram depicting the thresholded algorithm applied to an open double dot. The algorithm computes the continuous minimum (red cross) as in the default algorithm. However, rather than just considering all nearest neighbour discrete charge states, thresholding is used to reduce further the number of charge states needing to be evaluated compared to the default version of the algorithm. As the continuous minimum does not lie within the vertical grey rectangle, for which the width equals the threshold $t = 1/3$, it is unnecessary to consider both the floor and ceiling values for $N_1$. Instead, it can be rounded down. By contrast, the continuous minimum does lie in the horizontal rectangle, so both floor and ceiling values for $N_2$ must be considered. The discrete charge states, considered by the thresholded algorithm, are shown as red dots.

derived by neglecting constraints that the charge state must take integer, non-negative values, i.e. constraints (i) and (ii) respectively. If this solution satisfies constraint (ii), such that no dot contains a negative number of charges, we accept and use it. Otherwise, we compute a solution using a numerical solver, which explicitly applies constraint (ii). We try the analytical solution before resorting to the numerical solver because it is inexpensive to compute compared to the solver, and trying it first offers an overall time advantage.

### 1. The analytical solutions

The continuous minimum for an open quantum dot array is

$$\vec{Q}^{*\text{cont open}} = \mathbf{c}_{gd}\vec{V}_g, \tag{6}$$

see Appendix C for a formal derivation.

For closed quantum dot arrays, we use Lagrangian multipliers to account for the fixed number of charges in the array (imposed by constraint (iii)). As derived in Appendix D, the continuous solution can therefore be written as

$$\vec{Q}^{*\text{cont closed}} = \mathbf{c}_{gd}\vec{V}_g + \left(\hat{Q} - \vec{1}^T\mathbf{c}_{gd}\vec{V}_g\right)\frac{\mathbf{c}_{dd}\vec{1}}{\vec{1}^T\mathbf{c}_{dd}\vec{1}}, \tag{7}$$

where $\vec{1} = (1, 1, \ldots, 1)^T \in \mathbb{R}^{n_{\text{dot}}}$ is the one vector and $\hat{Q}$ is the confined charge.

In both the open and closed cases, one or more of the elements of $\vec{Q}^{*\text{cont open/closed}}$ may correspond to a negative number of electrons/holes. As mentioned before, if this occurs, we must fall back on the numerical solver to implicitly apply the constraint. The numerical solver is discussed next.

### 2. The numerical solver

The numerical solver computes the continuous minimum for open and closed dot arrays as a constrained convex quadratic optimisation problem. The general form of these problems is

$$\text{minimise } \vec{x}^T \mathbf{M} \vec{x} + \vec{v}^T \vec{x}$$
$$\text{subject to } \vec{l} \leq \mathbf{A}\vec{x} \leq \vec{u}. \tag{8}$$

In our case $\mathbf{M} = \mathbf{c}_{dd}^{-1}$ and $\vec{v} = -\mathbf{c}_{dd}^{-1}\mathbf{c}_{gd}\vec{V}_g$. Constraint (ii) can be encoded in this form by setting the constraint matrix $\mathbf{A}$ to the identity, and the lower and upper bound vectors ($\vec{l}$ and $\vec{u}$) to vectors of all zeros and infinities, respectively. For a closed array, constraint (iii) can be encoded by setting the constraint matrix as a row matrix with all terms equal to one and upper and lower bound vectors as single value elements equal to $\hat{N}$. Combining constraints for open and closed quantum dot arrays involves concatenating the associated constraint matrix and bound vectors.

## B. Evaluation of the nearest neighbour discrete charge states

With the continuous minimum in hand, the `QArray` algorithms iterate over the nearest neighbouring discrete charge states to find which one is the lowest in energy. The algorithms differ in how they define the nearest neighbouring discrete charge states. In the following subsection, we discuss precisely which charge states default and thresholded algorithms iterate over.

### 1. The default algorithm

The default algorithm will consider all the different charge configurations obtained by the element-wise flooring and ceiling of the values in the continuous minimum. Therefore, the default algorithm iterates over all $2^{n_{\text{dot}}}$ possible ways of rounding the elements of the continuous minimum up or down to the nearest integer. The charge states considered by the default algorithm when computing the lowest energy charge state of a double dot are highlighted in red in Fig. 2 (a). This considerably reduces the number of charge states that need to be considered compared to the brute force approach. As mentioned previously, we formally justify considering only the neighbouring discrete charge states in Appendix E. If the quantum dot array is closed, we eliminate the configurations that lead to the wrong number of charges in the array.

## 2. The thresholded algorithm

The motivation behind the thresholded algorithm was the observation that it was sometimes unnecessary to consider every one of the $2^{n_{\mathrm{dot}}}$ possible charge combinations. Only if the decimal part of each element of the continuous minimum charge state, $\vec{N}^{*\mathrm{cont}}$, is close to $1/2$ should we consider both charge configurations obtained by rounding up and down to the nearest integer. Otherwise, we can round to the closest charge configuration. As an example, the charge states considered by the thresholding algorithm when computing the lowest energy charge state of a double dot, with a threshold of $1/3$, are highlighted in red in Fig. 2 (b)).

If we define the threshold as $t \in [0,1]$, then the condition for having to consider both the floor and ceiling charge configurations on the $i$th quantum dot is

$$\left| \mathrm{frac}[N_i^{*\mathrm{cont}}] - 1/2 \right| \le t/2. \tag{9}$$

The total number of charge states needed to be evaluated after the thresholding scales with $(t+1)^{n_{\mathrm{dot}}}$ (see Appendix G for deviation).

When considering closed arrays, we found that applying the threshold may produce no charge states with the correct number of charges. In this case, we double the threshold and recompute. This threshold is left to the user's discretion; however, in Appendix F, we try to motivate a good choice of threshold.

## IV. ADDITIONAL FUNCTIONALITY

This section discusses features of `QArray` that are adjoint to the main algorithms. In particular, how we simulated charge sensing measurements (Sec. IV A) and thermal broadening (Sec. IV B), and present analytical results regarding optimal choice of gate voltages and virtual gates (Sec. IV C). These functionalities will be useful both when using `QArray` as a tool to understand charge stability diagrams and for automated tuning strategies. The charge sensing functions can be used to simulate realistic charge stability diagrams measured with a charge sensor. The optimal gate voltage function can be used to simulate a voltage sweep centred around a chosen charge state for any given capacitance matrix. Given a specific capacitance matrix, the optimal virtual gate function can be used to find the linear combination of gates that can modify the electrochemical potential of a single dot compensating for the effects of voltage cross-talk.

### A. Charge sensors

To simulate charge-sensing measurements, we distinguish between charge-sensing dots and regular dots. We first compute the charge state that minimises the free energy of the regular dots. With the charges on the regular dots fixed, we then determine the charge configuration on the sensor dots that minimises the overall free energy.

Next, we evaluate the difference in free energy between the minimum free energy charge state of the regular dots and the sensor's charge state next lowest in energy, which we denote as $\Delta F(\vec{V_g})$. We assume that the charge sensor dots are in the strongly coupled regime, so the conductance follows the Breit-Wigner formula [21]. Consequently, the charge sensor's

response is Lorentzian, and we model the sensor response as:

$$s(\vec{V}_g) = \frac{1}{(\Delta F(\vec{V}_g)/\Gamma)^2 + 1} \tag{10}$$

where $\Gamma$ is the sum of the tunnel rates to the source and drain; this parameter is configurable by the user to match their experimental setup. Additionally, we include various noise models to simulate white noise, $1/f$ noise, and sensor switches due to fluctuating charge traps, among others.

## B. Thermal Broadening

For a finite temperature, the quantum dot system will not adopt the lowest energy charge state. Instead, it will adopt a thermally mixed state based on the relative energy of the lowest energy charge states, resulting in the thermal broadening of charge transitions. We anticipate that the ability to simulate thermal effects would be useful for training representative data for neural networks. This can be accounted for by replacing the arg min function in Eq. (5) with a soft arg min function, which computes a Boltzmann weighted sum of the applicable charge states, such that

$$\vec{N}^* = \operatorname*{soft\,arg\,min}_{\vec{N} \in \mathcal{N}} F(\vec{N}; \vec{V}_g) \text{ , where}$$

$$\operatorname*{soft\,arg\,min}_{\vec{N}} F(\vec{N}; \vec{V}_g) = \frac{\sum_{\vec{N}} \vec{N} \exp[-F(\vec{N}; \vec{V}_g)/k_B T]}{\sum_{\vec{N}} \exp[-F(\vec{N}; \vec{V}_g)/k_B T]}. \tag{11}$$

The summation runs over all the allowed charge configurations considered by the algorithm. For a closed array, this means eliminating the charge states with the total number of charges different than $\hat{N}$. This approximation is valid for temperatures such that $k_B T$ is smaller than the dot charging energies, since the contribution of other charge states is exponentially suppressed.

## C. Optimal gate voltages and virtual gates

In developing `QArray`, we derived and implemented analytical results that help navigate charge stability diagrams. This section lists them.

*Optimal gate voltages*: the gate voltages that minimise the charge state $\vec{Q}_d$'s free energy, $F(\vec{V}_g, \vec{Q}_d)$ is

$$\vec{V}^* = (\mathbf{R}\mathbf{c}_{gd})^+ \mathbf{R}\vec{Q}_d, \tag{12}$$

where $\mathbf{R}^T\mathbf{R} = \mathbf{c}_{dd}^{-1}$ is the Cholesky factorisation and $+$ denotes the Moore-Penrose pseudo inverse. The proof of this result is presented in Appendix H.

*Optimal virtual gate matrix*: the optimal virtual gate matrix, $\alpha$, used to construct virtual gates is

$$\alpha = (\mathbf{c}_{dd}^{-1}\mathbf{c}_{gd})^+, \tag{13}$$

where $i$th row of this matrix encodes the electrostatic gate voltages required to change the $i$th dot's potential. The proof of this result is presented in Appendix I

## V. EXAMPLES

In this section, we explain the usage of `QArray` to produce the stability diagram of a double quantum dot. Firstly, we import the `DotArray` class as follows,

```
1  from qarray import DotArray, charge_state_changes
2  import numpy as np
3  import matplotlib.pyplot as plt
```

When initialising the `DotArray` class, we specify the system's capacitance matrices, charge carriers, the algorithm, and the implementation, as shown below. The temperature parameter ($T$) can be used to incorporate the effect of thermal broadening. In the example below, we set it to zero, ensuring that the solver determines the charge state that minimises the free energy as specified in Eq. (5).

```
1  # Initalising the DotArray class
2  array = DotArray(
3      Cdd=[
4          [0.0, 0.1],
5          [0.1, 0.0]
6      ],
7      Cgd=[
8          [1., 0.1],
9          [0.1, 1]
10     ],
11     algorithm = "default",
12     implementation = "rust",
13     charge_carrier = "holes",
14     T = 0.0
15 )
```

Here, the Cdd and Cgd arguments encode the dot-dot and dot-gate capacitive couplings, respectively. With the class' initialisation, these are stored as the attributes $c_{dd}$ and $c_{gd}$ (lowercase) in their Maxwell format. Once the model is defined, the user can either use a graphical user interface (GUI) or a programmatic one. We anticipate the GUI will be useful for fitting and understanding the charge stability diagrams of complex devices. Meanwhile, the programmatic interface is more flexible, allowing the generation of large datasets with access to the full breadth of QArray's functionalities, i.e. charge sensing, noise and thermal broadening. To use the GUI, the user should call the function:

```
1  array.run_gui()
```

This function opens a web interface on port 9000. Fig. 3 shows a screenshot of this interface. From the GUI, it is possible to configure parameter sweeps with voltage gates, virtual gates (denoted as $vP_i$), detuning (defined as $\epsilon_{ij} := vP_i - vP_j$) and onsite energy $U_{ij} := (vP_i - vP_j)/2$. As the sweep parameters are changed, the corresponding charge stability diagram is updated live, with the charge state being automatically labelled. In addition, it

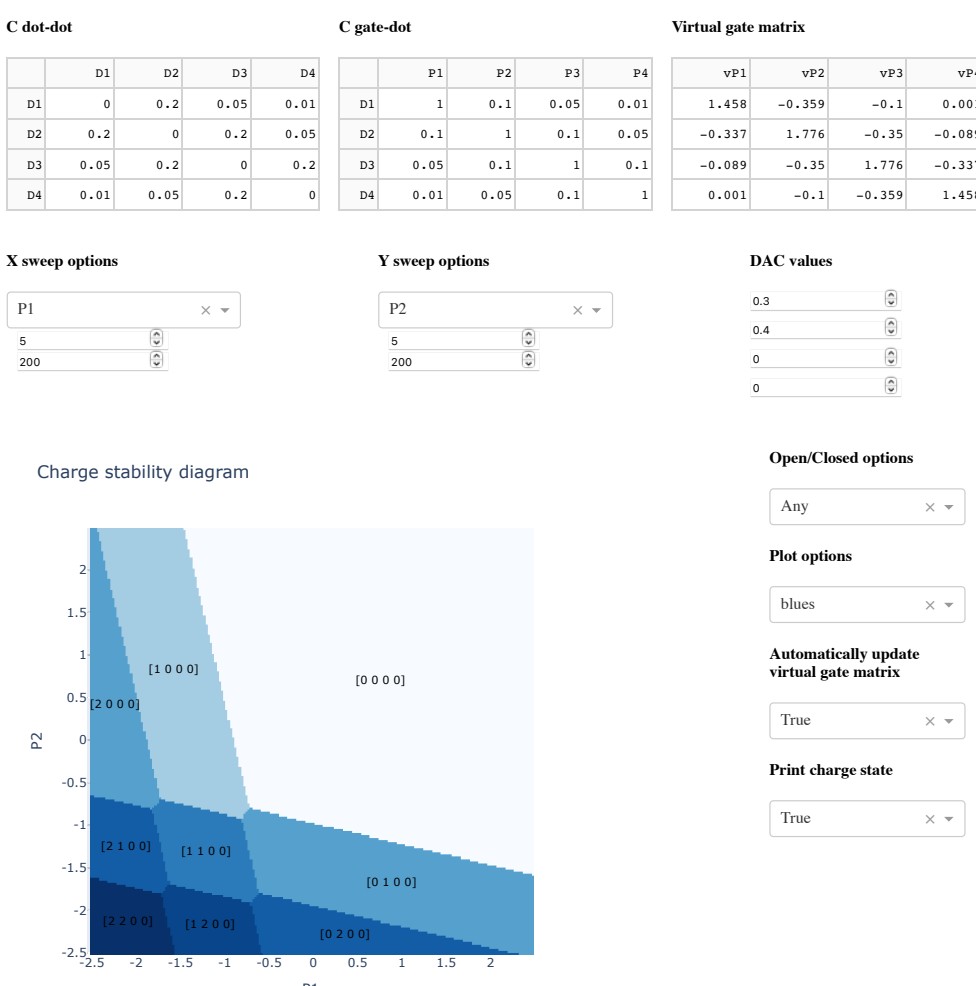

Figure 3. The QArray GUI. The matrices "C dot-dot" and "C gate-dot" can be used to configure the $C_{dd}$ and $C_{gd}$ capacitance matrices of a quantum dot array. The "virtual gate matrix" is used to configure the linear combinations of gates making up each virtual gate. This matrix can be updated automatically based on Eq. 13 or it can be edited manually (configured in the "automatically update virtual gate matrix" dropdown). The $x$ and $y$ sweep options and the "DAC values" can be used to perform any gate voltage sweep. The results are then plotted in the charge stability diagram plot section below. The open and closed drop down menu specifies whether the array is open or closed and, if it is closed, how many charges it contains. The plot options menu allows the user to choose the colour map, and finally, the print charge state menu allows the user to choose whether to label the corresponding charge state on the charge stability diagram. The plot is automatically updated whenever any of the options listed above are changed.

is possible to change the elements of the capacitance matrices and choose open or closed dot configurations.

The DotArray class has methods to perform one and two-dimensional gate voltage sweeps. For an open array, these methods are called do1d_open and do2d_open; likewise, do1d_closed and do2d_closed correspond to closed arrays. As shown below, these meth-

ods work with sweeps over arbitrary combinations of gate voltages, virtual plunger gates, detunings and onsite energies:

```
1  min, max, res = -4, 4, 400
2  # computing the ground state charge configurations
3  n = array.do2d_open("P1", min, max, res, "P2", min, max, res)
4  n_virtual = array.do2d_open("vP1", min, max, res, "vP2", min, max, res)
5  n_detuning_U = array.do2d_open("e1_2", min, max, res, "U1_2", min, max, res)
6  n_closed = array.do2d_closed("P1", min, max, res, "P2", min, max, res, n_charges = 2)
7
8  fig, ax = plt.subplots(2, 2, figsize=(5, 5))
9  extent = (min, max, min, max)
10 ax[0, 0].set_title("Open")
11 ax[0, 0].imshow(charge_state_changes(n), origin="lower", cmap='Greys', extent=extent)
12 ax[0, 0].set_xlabel("P1")
13 ax[0, 0].set_ylabel("P2")
14
15 ax[0, 1].set_title("Virtual gates")
16 ax[0, 1].imshow(charge_state_changes(n_virtual), origin="lower", cmap='Greys', extent =
       extent)
17 ax[0, 1].set_xlabel("vP1")
18 ax[0, 1].set_ylabel("vP2")
19
20 ax[1, 0].set_title("Detuning - on-site energy")
21 ax[1, 0].imshow(charge_state_changes(n_detuning_U), origin="lower", cmap='Greys', extent
       = extent)
22 ax[1, 0].set_xlabel("$e_{12}$")
23 ax[1, 0].set_ylabel("$U_{12}$")
24
25 ax[1, 1].set_title("Closed (2 holes)")
26 ax[1, 1].imshow(charge_state_changes(n_closed), origin="lower", cmap='Greys', extent =
       extent)
27 ax[1, 1].set_xlabel("P1")
28 ax[1, 1].set_ylabel("P2")
29 plt.show()
```

In this code, each of the do2d functions returns a numpy array of shapes (400, 400, 2) where the last axis encodes the number of holes in each dot for the corresponding gate voltages. To plot these charge configurations, we use charge_state_changes; this function checks whether the charge state changes in the $x$ or $y$ directions. Therefore, the function's output is a boolean array of shapes (399, 399) where 1s are present wherever charge transitions occur. QArray has other functions to map the charge states. The plots resulting from this code are shown in Fig. 4.

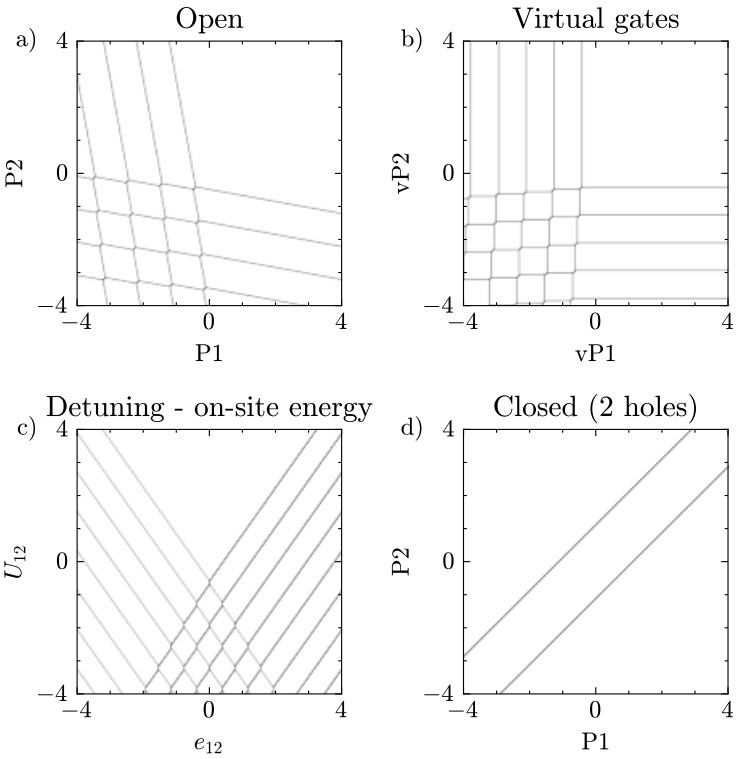

Figure 4. Plots from the code example. (a-c) show the charge stability diagrams of an open double dot obtained by sweeping the plunger gate voltages (a), a set of virtual gate voltages (b), and the detuning and on-site energies (c), respectively. (d) Charge stability diagram of thedouble dot considered in (a-c) in the close regime. Two holes are enclosed within the array.

## A. Realistic simulations

Beyond simply computing idealised charge stability diagrams `QArray` can simulate the response from a charge sensor and account for the thermal broadening of charge transitions, as explained in Section IV. Combined, these two capabilities make it possible to generate simulations of stability diagrams that look considerably closer to experimental data. Figure 5 (a) shows a charge sensor measurement of a charge stability diagram of an open quadruple dot presented in Ref. [22]. In Fig. 5 (b), we simulate this measurement using `QArray` with a $200 \times 200$ pixels resolution. Likewise, Fig. 5 (c) shows a measurement from Ref. [23] of a stability diagram corresponding to a five-electron closed configuration within a five-dot cross-geometry array. In Fig. 5 (d), we simulate this measurement with a resolution of $400 \times 400$ pixels. We achieved a very good qualitative agreement between both measurements and our simulations. Discrepancies are mainly due to the fact that the constant capacitance model does not capture the curvature of charge transitions; in this model, couplings between dots are kept constant for all gate voltage ranges.

The code to recreate the simulations displayed in Fig. 5 (b) and (d) is provided as examples within the `QArray` package. We used the Rust implementation of the default algorithm for both reconstructions, but we could have used any algorithm implementation. The compute times are listed in Appendix K. As a figure of merit, the Rust implementation of the default algorithm took 0.1 and 0.7s for the open and closed arrays, respectively. All

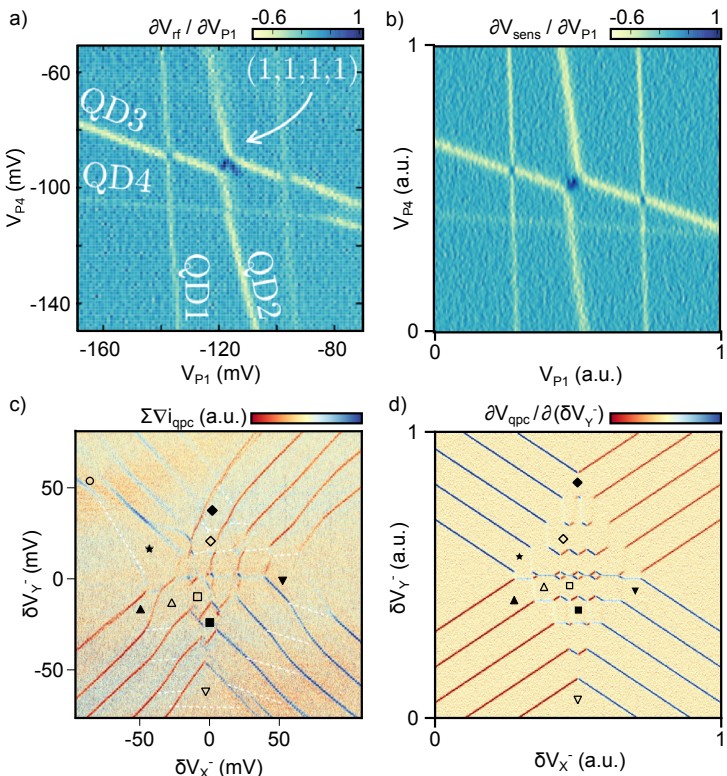

Figure 5. (a) Adaptation of Fig. 2b from Ref. [22] showing a stability diagram of a linear quadruple quantum dot array in the open regime. This is measured as a function of the two outermost plunger gates voltages, $V_{P1}$ and $V_{P4}$. The colour scale represents the derivative of the sensor signal $V_{rf}$ with respect to $V_{P1}$. (b) A recreation of Fig. 2b from Ref. [22], simulated using QArray. The colour scale represents the derivative of the simulated sensor signal, $V_{sens}$, with respect to $V_{P1}$. (c) Adaptation of Fig. 3a from Ref. [23] showing the stability diagram of a five-electron configuration corresponding to a five-dot cross-geometry array. The colour scale represents the sum of current changes in all charge sensors ($\sum \nabla i_{qpc}$) as a function of virtual gates $\delta V_X^-$ (controlling the left-right detunings) and $\delta V_Y^-$ (controlling the top-bottom detunings). (d) A recreation of Fig. 3a from [23] using QArray. The colour scale represents the derivative of the simulated sensor signal, in this case, $V_{qpc}$, with respect to $\delta V_Y^-$. To simulate the sensing of a QPC, we use the same method as in section IV A, using a linear profile instead of a Lorentzian one.

implementations of all algorithms in the package produce identical plots.

## VI. IMPLEMENTATIONS

Within `QArray`, the brute-force approach and our default algorithm are implemented in Python, Rust and JAX. The threshold algorithm is implemented in Python and Rust. The Rust and JAX implementations fulfil different use cases. The Rust implementations are optimised for computation on a CPU, while a GPU can accelerate the JAX-based implementations. The Python implementation does not provide a practical advantage over the Rust and JAX cores, but it is retained for benchmarking purposes (see Fig. 6(a-b)). In this section, we discuss these different implementations in detail.

*The Rust implementations*: Rust is a systems programming language on par with `c` in speed [13, 24, 25]. Our Rust implementations take advantage of parallelism when sensible, based on workload at runtime, thanks to the functionality provided by `rayon` [13, 14, 24, 25]. In addition, we used caching and function memorisation to avoid reevaluating the discrete charge state configurations for which the free energy is evaluated. However, appreciating that Rust is a relatively niche language compared to the ubiquitousness of Python, we interloped the Rust core with Python. This allows the user to gain all the benefits of Rust in their familiar Python coding environment.

*The JAX implementations*: JAX is a Python-based machine learning framework that combines autograd and XLA (accelerated linear algebra). Its structure and workflow mirror that of `numpy`; however, at run time, the code can be just in time (`jit`) compiled and auto-vectorised to primitive operations for significant performance improvements. Through the `vmap` function, JAX will compile the functions to XLA and execute them in parallel with a GPU [15]. Unfortunately, the discrete nature of the constant capacitance model negates the possibility of using JAX's autograd capabilities. In addition, as the sizes of all arrays must be known at just-in-time compile time, the JAX implementation of our algorithm cannot perform the thresholding algorithm.

*The numerical solver*: We made use of the OSQP (Operator Splitting Quadratic Program) solver instead of relying on less specific optimisers available in `scipy.optimize` [26]. The OSQP solver is optimised explicitly for constrained convex quadratic problems [27]. It is an incredibly efficient solver; only one matrix factorisation is required to set up the optimisation problem, after which all operations are cheap, such that there is no need to perform any slow division operations. For a complete description of the OSQP solver and its benchmarks, see Ref. [27]. For the solver's hyperparameters, we use the default values suggested by this reference. As a result, the solver performs the required number of iterations to obtain the continuous minimum to within an absolute and relative error of $10^{-3}$. The solver updates the step size automatically. Our benchmarks found that the OSQP solver was able to find the continuous minimum of double, triple and even quadruple dot systems in less than $10\,\mu$s per voltage configuration.

It is important to note that in `QArray`, we set the unit charge, $|e| = 1$, to avoid issues with numerical stability. The units of free energy are thus electron volts, and the units of our temperature parameter are Kelvin.

## VII. BENCHMARKS

We benchmark the performance of each of the algorithms within `QArray` for all the different implementations discussed. In order to do this, we generate a random $C$ matrix by drawing each element from a uniform distribution. This randomised matrix is converted to the Maxwell format. We then compute the ground state charge configuration over a set of $100 \times 100$ randomly chosen gate voltage configurations and record the average computation time.

For all the implementations of the brute-force and default algorithm, we show the average compute time over many benchmarking runs executed on both the CPU within the Apple M1 Pro System on Chip (SOC) and an NVIDIA GTX 1080TI GPU (Fig. 6 (a-b)). As anticipated, the default version of the algorithm exhibits superior scalability compared to the brute force approach, resulting in notably shallower performance curves. Compared to the brute force method, for arrays with more than five dots, the Rust and JAX implementations

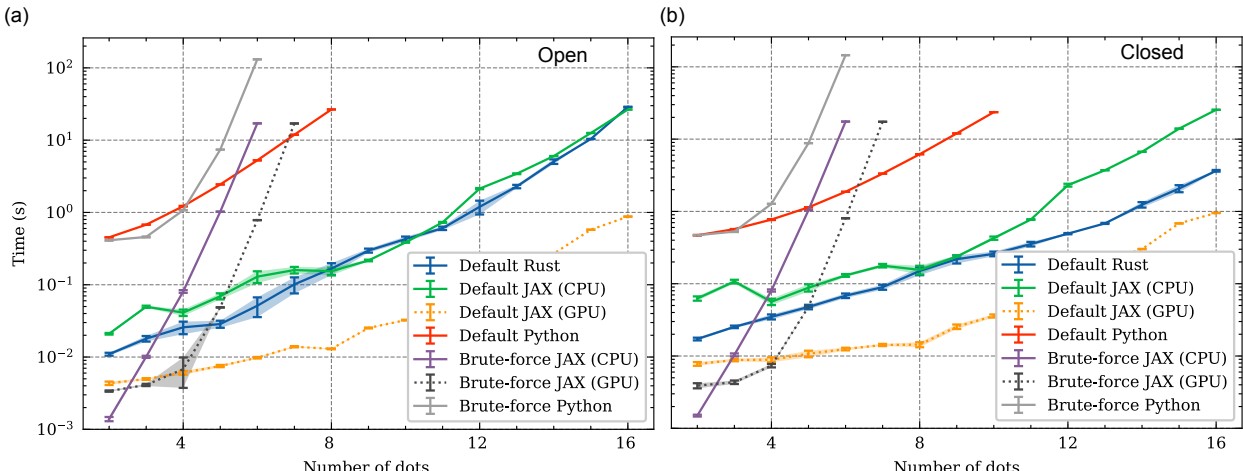

Figure 6. (a-b) Average computation time required by each core to generate $100 \times 100$ pixels charge stability diagrams. Simulations are performed for randomly chosen capacitance matrices and gate voltage configurations and correspond to increasing sizes of both open (top) and closed (bottom) arrays. We ran the JAX and brute force cores on both a CPU and a GPU. For the closed quantum dot arrays, the number of confined electrons/holes matches the number of quantum dots within the array. The error bars represent confidence intervals of $2\sigma$, such that for a randomly chosen set of capacitance matrices and voltages, the compute time should fall within the error bars 95% of the time.

of the default algorithm are at least one order of magnitude faster and gain more time advantage as the number of dots increases. Even the Python implementation becomes faster than the GPU-accelerated brute force method for arrays with more than six quantum dots. Still, it remains several orders of magnitude slower than the Rust and JAX implementations due to the speed gained by the parallelisation and the GPU usage offered by these cores. On the CPU, our Rust and JAX cores demonstrate approximately equal performances for open dot arrays. However, for closed dot arrays, the Rust core leverages optimisations that confer enhanced performance in the simulation of large arrays. Nevertheless, the advantage of these optimisations is eclipsed by the raw computational power harnessed by the JAX core when executed on the GPU. The GPU-accelerated JAX algorithm can compute the charge stability diagram of a 16-dot array featuring $100 \times 100$ gate voltage pixels in less than a second for both open and closed configurations.

In Fig. 7 (a-b), we compare several runs obtained with the Rust implementation of the thresholded algorithm using different values of $t$ for open and closed regimes. As expected, smaller threshold values result in faster computation, especially in larger arrays. The Rust thresholded algorithm running on a CPU was faster than the JAX implementation on a GPU for $t < 2/3$. We also note larger time gains in the open dots configuration compared to the closed case. This difference arises because when simulating quantum dot arrays in the closed regime, the threshold strategy may fail if no charge state configurations with the correct number of charges are found. In this case, the algorithm doubles the threshold value and recomputes the stability diagram. While this method ensures a correct stability diagram is generated, the additional computational overhead leads to smaller time gains than in the open dot case.

Whilst decreasing the $t$ values yields dramatic performance gains, it should not be reduced

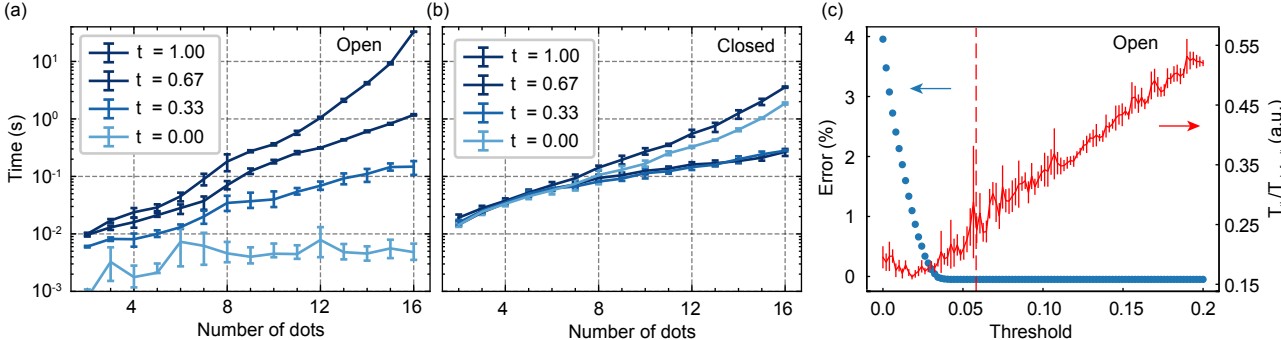

Figure 7. (a-b) Average computation time required by the Rust core to generate a $100 \times 100$ pixels charge stability diagram computed with different threshold values for increasing numbers of quantum dots both for open and closed quantum dot arrays. The error bars denote the two standard deviation confidence intervals. (c) Comparison between the pixel values corresponding to the charge stability diagram of Fig. 5(b) computed with and without the threshold algorithm. The axis on the left is the percentage error obtained for the thresholded algorithm as a function of the threshold value. The axis on the right is the time saved when using the threshold algorithm. The error bars represent the standard deviation of the calculation time over 10 repetitions. The dashed line corresponds to an optimal threshold value that achieves a time save of a factor of 4 without introducing artefacts in the charge stability diagram.

too much as this might introduce artefacts in the simulated charge stability diagrams (see Appendix J). To evaluate the error that could be introduced by the threshold method, we compute the charge stability diagram of Fig. 5(b) with different threshold values between 0 and 0.2 and compare it with the stability diagram simulated with our default algorithm. The axis on the left of Fig. 7(c) is an estimated percentage error as a function of the threshold value. The error is obtained from a pixel by pixel comparison of two charge stability diagrams (with and without thresholds), only considering the pixels with a discrepancy larger than the numerical precision. In this specific example, we find that for the lowest value of the threshold, the maximum error is about 4%. On the right axis of the same plot, we report the fraction of the time taken by the threshold algorithm ($T_{th}$) against the standard algorithm ($T_{default}$). In Appendix F, we provide an analytical justification for selecting the threshold based on the parameters of the capacitance matrices. The dashed line indicates a good threshold value, as given by Eq. F3 in the appendix, which leads to a 4 times computational speedup without errors compared to the standard algorithm. The error bars indicate the standard deviation of the computational time over 10 repetitions of the charge stability diagram calculations.

We note that the minimum threshold that can be set without introducing artefacts depends on the $\mathbf{c}_{dd}^{-1}$ capacitance matrix. See Appendix E and F for a detailed discussion. In a few words, we have found that the minimum threshold is typically of the order of the ratio of the largest off-diagonal element of the $\mathbf{c}_{dd}$ capacitance matrix to its corresponding diagonal element. Given that the diagonal elements are the dot's total capacitive coupling to every other dot and gate, whilst the off-diagonal elements are dot-dot couplings, we find the critical threshold to be much smaller than one in the case of weakly coupled dots, the approximation under which the capacitance model is justified. Therefore, the performance shown in Fig. 7 should be interpreted as the performance gained without introducing dis-

tortions if the quantum dot array's capacitance matrices allows setting a sufficiently low threshold.

We note that these benchmarks, based on randomised gate voltages and capacitance matrices, might underestimate the potential of the thresholded algorithm. In tuning larger quantum dot arrays, virtual gates are often used to control isolated double dot subsystems whilst the other dots remain unperturbed (the $n+1$ method [28]). In this case, the threshold algorithm could neglect the charge states corresponding to the other dots. Therefore, the time required to simulate a large array operated in this way would be comparable to simulating a double dot device.

## VIII.   SUMMARY AND OUTLOOK

We demonstrated that our open-source software package can accelerate the simulation of charge stability diagrams of large quantum dot arrays in both open and closed regimes. The speed and GPU acceleration will aid in generating larger, more diverse datasets on which to train neural networks. This will improve the accuracy of the neural network-based classifiers used in automatic tuning approaches. With improved intuition about the charge stability diagrams, new tuning methods might be developed. In particular, the automated tuning of closed arrays could be drastically easier than for open arrays owing to the smaller number of transitions. The speed of our algorithms is particularly promising for model-based machine-learning methods and real-time interfacing with experiments, as well as for the development of hardware-in-the-loop approaches.

In future, we hope that, with help from the wider community, this package can grow to include more advanced noise models and high-level functionality.

## CODE AVAILABILITY

The code is available on the Python package index under `QArray`, so is pip installable with the command `pip install QArray`. The associated GitHub repositories are `https://github.com/b-vanstraaten/QArray`. Please star the repository and cite this paper if this package is useful. Documentation is available at `https://QArray.readthedocs.io/en/latest/introduction.html`. Discovered bugs can be reported using GitHub issues.

## AUTHOR CONTRIBUTIONS

B.v.S. wrote the code. B.v.S. and J.D.H wrote the documentation. B.v.S. and L.S. developed the mathematical proofs and derived the analytical results. B.v.S., J.D.H. and N.A. conceived of creating an open-source capacitance model software package. All authors contributed to the manuscript.

## ACKNOWLEDGEMENTS

This work was supported by the Royal Society, the EPSRC Platform Grant (EP/R029229/1), and the European Research Council (Grant agreement 948932).

## COMPETING INTERESTS

Natalia Ares declares a competing interest as a founder of QuantrolOx, which develops machine learning-based software for quantum control.

---

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

## Appendix A: Software implementations

This section lists the software implementations of each algorithm and how many charge states they consider.

| Algorithm | Number of charge states to be considered | Software implementations |
|---|---|---|
| Brute-force | $(n_{\max} + 1)^{n_{\mathrm{dot}}}$ where $n_{\max} \in \{1, 2, ..., \infty\}$ | JAX, Python |
| `QArray`-default | $2^{n_{\mathrm{dot}}}$ | Rust, JAX, Python |
| `QArray`-thresholded | $(t + 1)^{n_{\mathrm{dot}}}$ where $t \in [0, 1]$ | Rust, Python |

Table I. A table showing the number of charge states needing to be considered by the software implementations of the brute-force, default and thresholded algorithms. Where $n_{\mathrm{dot}}$ refers to the number of quantum dots in the array being simulated, is $n_{\max}$ maximum number of charges considered in any dot by the brute-force algorithm and $t$ is the threshold used in the thresholded algorithms, which quantifies the degree of approximation.

## Appendix B: Positive Definite Proof

This section proves that the $\mathbf{c}_{dd}$ is positive definite, making the optimisation problem convex.

**Theorem B.1** *For any set of capacitances between nodes such that $c_{ij} \, \forall i, j$ are real, non-negative and $c_{ij} = c_{ji}$ the Maxwell matrix $\mathbf{C}^M \in \mathbb{R}^{(n_{dot}+n_{gate}) \times (n_{dot}+n_{gate})}$ is defined as*

$$C_{ij}^M = \left( \sum_k c_{ik} \right) \delta_{ij} - c_{ij}(1 - \delta_{ij}). \tag{B1}$$

*Let $\mathbf{C} \in \mathbb{R}^{n_{dot} \times n_{dot}}$ be the upper left block of $\mathbf{C}^M$. $\mathbf{C}$ is positive definite assuming the physical case of every dot being coupled to at least one gate, namely $\forall i \, \exists j > n_{dot}$ such that $c_{ij} > 0$.*

**Proof B.2** *A matrix $\mathbf{A} \in \mathbb{R}^{n \times n}$ is called strictly diagonally dominant if for all $n$ diagonal entries $|A_{ii}| > \sum_{k \neq i} |A_{ik}|$ holds. We start proving that $\mathbf{C}$ is strictly diagonally dominant by observing*

$$C_{ii} = \sum_{k=1}^{n_{dot}+n_{gate}} c_{ik} > \sum_{k \neq i}^{n_{dot}} |C_{ik}| = \sum_{k \neq i}^{n_{dot}} c_{ik}.$$

*This holds due to the non-negative elements $c_{ij}$ and can be simplified to the true statement $\sum_{k=n_{dot}+1}^{n_{dot}+n_{gate}} c_{ik} > 0$. Strict diagonal dominance implies positive definiteness for symmetric matrices with non-negative elements by [29, Theorem 6.1.10].*

## Appendix C: Continous minimum for open quantum dot arrays

In this section, we derive the continuous minimum charge state for open quantum dot array - the charge state that minimises (4) neglecting constraints (i) and (ii). The free energy is

$$F(\vec{V}_g, \vec{Q}_d) := \frac{1}{2}(\mathbf{c}_{dg}\vec{V}_g - \vec{Q}_d)^T \mathbf{c}_{dd}^{-1}(C_{dg}\vec{V}_g - \vec{Q}_d) \tag{C1}$$

Where $\vec{V}_g$ denotes the applied gate voltages, $\vec{Q}_d$ the charge on the dots and $c_{dd}$ and $c_{dg}$ are submatrices of the Maxwell capacitance matrix, see Sec.II). For ease of notation going forward, we will write this as

$$F(\vec{V}, \vec{Q}) = \frac{1}{2}(V - Q)^T C^{-1}(V - Q) \tag{C2}$$

where $\vec{V} := \mathbf{c}_{gd}\vec{V}_d$. Differentiating $F$ for $Q_i$ gives

$$\frac{\partial F}{\partial Q_i} = -\left[C^{-1}(V - Q))\right]_i \overset{!}{=} 0 \longrightarrow Q_i = V_i \tag{C3}$$

Therefore, the continuous minimum for an open quantum dot array is

$$\vec{Q}^{*\text{cont open}} = \vec{V} \tag{C4}$$

**Appendix D: Continous minimum for closed quantum dot arrays**

In this section, we derive the continuous minimum charge state for closed quantum dot arrays - the charge state that minimises (4) neglecting constraints (i) and (ii). If the array contains $\hat{Q}$ charges, we can incorporate (iii) through the method of Lagrangian multipliers with the Lagrangian

$$L(\vec{V}_g, \vec{Q}_d, \hat{N}) := \frac{1}{2}(\mathbf{c}_{dg}\vec{V}_g - \vec{Q}_d)^T\mathbf{c}_{dd}^{-1}(C_{dg}\vec{V}_g - \vec{Q}_d) - \lambda\left(\sum_i Q_{di} - \hat{Q}\right). \tag{D1}$$

For ease of notation going forward, we will write this as

$$L(\vec{V}, \vec{Q}, \hat{Q}) = \frac{1}{2}(V - Q)^T C^{-1}(V - Q) - \lambda\left(Q^T\vec{1} - \hat{Q}\right) \tag{D2}$$

where $\vec{V} := \mathbf{c}_{gd}\vec{V}_d$, whilst $\vec{1}$ is the one vector with all entries one, and otherwise subscripts have been dropped. Differentiating $L$ with respect to $\lambda$ yields

$$\frac{\partial L}{\partial \lambda} = -Q^T\vec{1} + \hat{Q} \overset{!}{=} 0 \to \hat{Q} = Q^T\vec{1}. \tag{D3}$$

Differentiating $L$ for $Q_i$ gives

$$\frac{\partial L}{\partial Q_i} = -\left[C^{-1}(V - Q))\right]_i - \lambda \overset{!}{=} 0 \longrightarrow Q_i = V_i + \lambda C\vec{1}, \tag{D4}$$

Plugging (D4) into (D3) and solving for $\lambda$ yields

$$\lambda^* = \frac{\hat{Q} - \vec{1}^T V}{\vec{1}^T C\vec{1}}. \tag{D5}$$

The Lagrangian in (D2) can be rewritten as

$$\begin{aligned}
L(\vec{V}, \vec{Q}, \hat{Q}) &= \frac{1}{2}Q^T C^{-1}Q - V^T C^{-1}Q + \frac{1}{2}V^T C^{-1}V - \lambda^*\vec{1}^T Q + \lambda^*\hat{Q} \\
&= \frac{1}{2}\left(V + C\vec{1}\lambda^* - Q\right)^T C^{-1}\left(V + C\vec{1}\lambda^* - Q\right) + \lambda^*\hat{Q} - \lambda^* V^T\vec{1} - \frac{1}{2}\lambda^{*2}\vec{1}^T C\vec{1}
\end{aligned} \tag{D6}$$

by using the matrix generalisation of completing the square. Since the last three terms do not depend on $\vec{Q}$, this new Lagrangian is again quadratic in $\vec{Q}$ with the positive definite matrix $C$. Therefore, the continuous minimum is

$$Q^{*\text{cont min}} = V + C\vec{1}\lambda^*$$

$$= V + \left(\hat{Q} - \vec{1}^T\vec{V}\right)\frac{C\vec{1}}{\vec{1}^TC\vec{1}}. \tag{D7}$$

**Appendix E: Proofs of Optimality for Considering Surrounding Neighbours and Thresholding**

This section discusses whether only considering the nearest charge states to the continuous minimum is sufficient. We consider two limiting cases and prove that in these cases, it is sufficient. The optimisation problem set out in equation (5) can be written in the form

$$\arg\min_{\vec{x}\in\mathbb{R}^n} \frac{1}{2}\vec{x}^T\mathbf{M}\vec{x} + \vec{b}^T\vec{x} \tag{E1}$$

$$\text{subject to } x_i \geq 0 \ \forall i \tag{E2}$$

$$\text{and } x_i \in \mathbb{Z} \ \forall i, \tag{E3}$$

where $\mathbf{M}$ is positive definite matrix. In the following, we prove that if $M$ were diagonal, it would be sufficient to always round the continuous minimum to the nearest discrete change state.

**Theorem E.1** *The global minimiser $\vec{x}^*_{discret}$ of the optimisation problem (E1),(E2), and (E3) can be obtained by rounding each component of the minimiser $\vec{x}^*_{cont}$ to the continuous optimisation problem consisting of (E1) and (E2).*

**Proof E.2** *Since $\mathbf{M}$ is diagonal, the optimisation problem uncouples to $n$ independent one-dimensional optimisation problems of the form $\min_{x_i} 1/2M_{ii}x_i^2 + b_ix_i$ with the non-negativity constraint $x_i \geq 0$ and the integer constraint $x_i \in \mathbb{Z}$. Now we show that rounding the solution $x^*_{i,cont}$ from the one-dimensional continuous optimisation problem to the closest integer yields the global minimiser $x^*_{i,discret}$. If $x^*_{i,cont} = 0$, the integer constraint is already fulfilled. If $x^*_{i,cont} > 0$, then the parabola is symmetric around $x^*_{i,cont}$. Therefore, rounding to the closest integer minimises the one-dimensional integer-constrained optimisation problem.*

With small perturbations away from this diagonal case, we cannot always round to the nearest charge to reliably find the lowest energy configuration. Nevertheless, we can guarantee that the minimiser is one of the $2^{nn}$ surrounding neighbours around the continuous solution as long as the condition number $\kappa(\mathbf{M}) = \lambda_{max}/\lambda_{min}$ is sufficiently low. intuitively, a low condition number corresponds to an almost spherical symmetric potential.

**Theorem E.3** *If $\kappa(\mathbf{M}) \leq 1 + 4/n$, then one of the $2^n$ surrounding neighbours of the continuous minimiser $x^*_{cont}$ is the global minimiser $x^*_{discret}$ of the integer optimisation problem.*

**Proof E.4** *We proceed by showing that even the closest integer point apart from the surrounding neighbours of the continuous minimiser has a larger objective function value than*

*one of the surrounding neighbours for any matrix* $\mathbf{M}$ *as long as the condition number is sufficiently low. Let* $\vec{a}$ *be the vector from* $\vec{x}^*_{cont}$ *to the closest integer neighbour and* $\vec{b}$ *the vector from* $\vec{x}^*_{cont}$ *to the closest integer point which is not a surrounding neighbour of* $\vec{x}^*_{cont}$*. Without loss of generality, we can assume that* $\vec{a}$ *and* $\vec{b}$ *have the form* $\vec{a} = [\vec{c}; d]$ *and* $\vec{b} = [\vec{c}; d+1]$*. ( The notation means that* $\vec{a}$ *has the same entries as* $\vec{c}$ *with an additional appended entry d.)*

*Let* $g(\vec{y}) = \vec{y}^T \mathbf{M} \vec{y}/2$ *with* $\vec{y} = \vec{x} - \vec{x}^*_{cont}$ *such that* $||\vec{y}||^2 \lambda_{min} \leq g(\vec{y}) \leq ||\vec{y}||^2 \lambda_{max}$ *holds* $\forall \vec{y}$*. We want to show that* $g(\vec{a}) \leq g(\vec{b})$ *which is necessarily true if* $||\vec{a}||^2 \lambda_{max} \leq ||\vec{b}||^2 \lambda_{min}$*. This yields*

$$\frac{\lambda_{max}}{\lambda_{min}} \leq \frac{||\vec{c}||^2 + (d+1)^2}{||\vec{c}||^2 + d^2}. \tag{E4}$$

*It holds that* $||\vec{c}||^2 \leq (n-1)/4$ *since* $||\vec{a}||^2$ *is defined to be the vector to the closest neighbour and therefore* $|a_i| \leq 1/2 \, \forall i$*. By incorporating this bound, we obtain the desired result of* $\lambda_{max}/\lambda_{min} \leq 1 + 4/n$*.*

We showed that under certain conditions, the discrete minimiser is among the surrounding neighbours. Intuitively we can do better, namely, if the continuous minimiser is close to an integer in one variable, then the discrete minimiser can be obtained by rounding to the closest integer. The following theorem derives bounds when rounding single coordinates leads to optimal solutions.

**Theorem E.5** *Let element* $i$ *of* $\vec{x}^*_{cont}$ *be in the interval* $[k, k+\delta]$ *for an integer* $k$*. If the condition number* $\kappa$ *fulfills*

$$\delta \leq \frac{\sqrt{\kappa^2 - (n-1)(\kappa^2 - 1)^2} - 1}{\kappa^2 - 1}, \tag{E5}$$

*the* $i$*-th element of the discrete minimiser is* $k$*.*

**Proof E.6** *We proceed in a similar way as beforehand by showing that no integer point with* $k$ *at its* $i$*-th element has a higher value than the "opposite" point with entry* $k+1$*. With no loss of generality, let* $\vec{a} = [\vec{c}; x]$ *and* $\vec{b} = [\vec{c}; 1-x]$ *be the vectors from the continuous minimiser to two opposite integer points. Plugging this into* $||\vec{a}||^2 \lambda_{max} \leq ||\vec{b}||^2 \lambda_{min}$ *yields*

$$\frac{\lambda_{max}}{\lambda_{min}} \leq \sqrt{\frac{||\vec{c}||^2 + (1-\delta)^2}{||\vec{c}||^2 + \delta^2}}. \tag{E6}$$

*Rearranging yields the quadratic* $(\kappa^2 - 1)\delta^2 + 2\delta - 1 + (\kappa^2 - 1)||\vec{c}||^2 \leq 0$*. Solving that for* $\delta$ *and observing that* $||\vec{c}||^2 \leq n-1$ *yields the desired result.*

Under this condition, we can guarantee that certain surrounding nodes can be omitted from checking and can, therefore, choose a threshold of $t = 1 - 2\delta$. Experimentally, however, choosing lower thresholds still yields optimal results, e.g. $t = ||\mathbf{\Delta}||_2$ - we motivate this choice in the following section.

**Appendix F: Physical motivation for diagonal and spherical limit case**

The constant interaction model is a good approximation when the quantum dots interact weakly, and the associated tunnel coupling is small. As a result, we should expect the interdot capacitive coupling to be smaller than the coupling to the nearest gates; the diagonal elements will be considerably larger than the off-diagonals. As a result, we can write

$$\mathbf{c}_{dd} = \mathbf{D}(\mathbf{I} - \mathbf{\Delta}) \tag{F1}$$

where $\mathbf{D}$ is a diagonal matrix, $\mathbf{I}$ is the identity matrix, and the elements $\mathbf{\Delta}$ are given by the ratio of the off-diagonal terms to the on. The inverse of this matrix can be approximated using the Neumann expansion

$$\mathbf{c}_{dd}^{-1} = \sum_{n=0}^{\infty} \mathbf{\Delta}^n \mathbf{D}^{-1} \approx (\mathbf{I} + \mathbf{\Delta})\mathbf{D}^{-1}. \tag{F2}$$

Therefore, for small $\mathbf{\Delta}$ where this approximation is valid, the $\mathbf{c}_{dd}^{-1}$ matrix is diagonally dominant and is a perturbation from the diagonal matrix. If $\mathbf{c}_{dd}$ were diagonal, the charges in the quantum dots would not interact. In this case, the lowest energy discrete charge configuration could be found by rounding the continuous solution to the nearest integer charge, as we prove in the previous section.

It follows that for small but non-zero $\mathbf{\Delta}$ rounding will yield the lowest energy charge state except in the most ambiguous cases where a fractional component of the ith element of the continuous minimum, $N_i^*$, is close to $1/2$. Therefore, the threshold should be proportional to some $\mathbf{\Delta}$ norm, for example

$$t/2 = ||\mathbf{\Delta}||_2. \tag{F3}$$

**Appendix G: Number of charge states**

**Theorem G.1** *The expected number of charge states needing to be considered by the thresholded algorithm with a threshold $t$ is given by $(1+t)^{n_{dot}}$ where $n_{dot}$ is the number of quantum dots.*

**Proof G.2** *For random uniformly distributed gate voltages, the continuous minimum can also be considered uniformly distributed. Therefore, in 1d, the probability that the algorithm has to consider one charge state is $1 - t$, and the probability that it has to consider two is $t$. Therefore, the expected number of points is $\mathbb{E}[N_{1d}] = t \cdot 2 + (1 - t) \cdot 1 = 1 + t$. As the dimensions are independent $\mathbb{E}[N_{Nd}] = \mathbb{E}[N_{1d}]^N$. Therefore, if there are $n_{dot}$ quantum dots the expected number of charge states is $(1 + t)^{n_{dot}}$*

**Appendix H: Finding optimal voltage for a given charge state**

**Theorem H.1** *The gate voltages that minimise the charge state $\vec{Q}_d$ free energy, $F(\vec{V}_g, \vec{Q}_d)$ is*

$$\vec{V}^* = (\mathbf{R}\mathbf{c}_{gd})^+ \mathbf{R}\vec{Q}_d, \tag{H1}$$

*where $\mathbf{R}^T\mathbf{R} = \mathbf{c}_{dd}^{-1}$ is the Cholesky factorisation and $+$ denotes the Moore-Penrose pseudo inverse.*

**Proof H.2** *The free energy is given by*

$$F(\vec{V_g}, \vec{N_d}) = \frac{1}{2} \left( \vec{Q_d} - \mathbf{c}_{gd}\vec{V_g} \right)^T \mathbf{C}^{-1} \left( \vec{Q_d} - \mathbf{c}_{gd}\vec{V_g} \right). \tag{H2}$$

*Using the Cholesky factorisation $\mathbf{c}_{dd}^{-1} = R^T R$, the minimisation problem can be reformulated as the following linear least-squares problem*

$$\min_{\vec{V_g}} ||\mathbf{R}\vec{Q_d} - \mathbf{R}\mathbf{c}_{gd}\vec{V_g}||_2. \tag{H3}$$

*Solving this via the Moore-Penrose pseudo inverse yields the desired result.*

### Appendix I: Virtual gates

This section derives how to construct virtual gates, namely determining how to change the gate voltages to obtain a specific change in the dot potentials.

**Theorem I.1** *The optimal virtual gate matrix, $\alpha$, used to construct virtual gates is*

$$\alpha = (\mathbf{c}_{gd}\mathbf{c}_{dd}^{-1})^+ \tag{I1}$$

*where $+$ denotes the Moore-Penrose pseudoinverse. The ith row of this matrix encodes the electrostatic gate voltages required to change the ith dot's potential.*

**Proof I.2** *The quantum dot potential changes with gate voltages according to $\Delta\vec{V_d} = \mathbf{c}_{dd}^{-1}\mathbf{c}_{gd}\Delta\vec{V_g}$ according to (3). If this simple linear system is invertible, the Moore-Penrose pseudoinverse boils down to the regular inverse. If it is underdetermined (i.e., more gates than dots and full row rank), there are infinite solutions. In this case, the pseudoinverse picks the solution with minimal 2-norm. In case of an overdetermined linear system (i.e. more dots than gates or no full column rank) there is no solution. In this case, the pseudoinverse returns the least squares approximate solution.*

### Appendix J: Effect of threshold

As evidenced in Figure 7, the thresholding strategy can yield significant reductions in the compute time by reducing the number of charge states evaluated to determine whether they are the lowest in energy. However, using values that are too small for the threshold will introduce artefacts in the charge stability diagram. Decreasing the threshold further will exaggerate these artefacts to the degree that for $t = 0$, all interdot transitions will be lost. We demonstrate this in Fig. 8 a), where we simulate a double quantum dot with dot-dot capacitance matrix

$$c_{dd} = \begin{bmatrix} 1.4 & -0.2 \\ -0.2 & 1.4 \end{bmatrix}. \tag{J1}$$

In this case, our empirical understanding of the threshold suggests the minimum value should be the ratio of the off-diagonal elements to the on so $t_{\min} = 0.2/1.4 = 1/7$, this is confirmed by plots e) and f) in Figure 8 being identical. For smaller but non-zero values of the threshold, the charge stability diagram is distorted with the size of the interdot transition

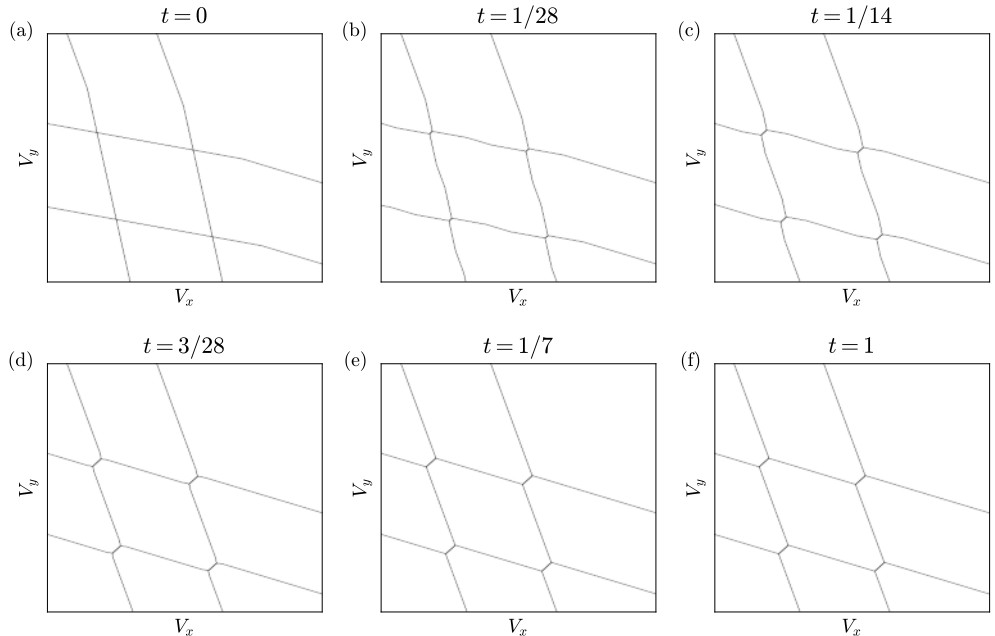

Figure 8.      (a-e) Charge stability diagrams of an open double quantum dot produced by the thresholded algorithm for threshold values smaller than or equal to the empirical minimum value of 1/7 based on the capacitance matrix. The value of the threshold used in the simulation is stated above in the plot. f) Simulated charge stability diagram with the thresholded algorithm with the threshold set to one, so it performs identically to the default algorithm.

being suppressed with decreasing threshold (Fig. 8 b-d)). In Appendix F, we motivate a formula for the minimum value of the threshold $t$. In this case, the value suggested is more conservative at 0.202. While not wrong, this value is quite a bit larger than what empirical intuition suggests; more work is required to understand the thresholded algorithm fully.

In addition, even when the threshold is well above the minimum, the thresholding strategy appears to introduce almost imperceptible distortions into the charge stability diagram. Figure 9 compares the charge stability diagram produced by the Rust implementation threshold set to 1/3 with that produced by the brute force implemented in JAX for a quadruple dot. For the capacitance matrix used in this simulation, our empirical understanding of the threshold value required to capture the interdot transitions accurately is $t_{\min} = 0.08$, whilst the analytical formula gives 0.19. We hypothesise finite precision of the OSQP solver, which is exacerbated by the thresholding. For example, if a threshold of $t = 2/3$ is used, the charge stability diagram is identical to brute force.

In summary, more work is required to understand the implications of the thresholding strategy fully. However, it is worth noting that these distortions in the charge stability diagram are next to irrelevant when using the constant capacitance model to generate training data for neural networks such as in references [9, 10]. As effects we manually add to make the charge stability diagram look more , such as thermal broadening, measuring the charge stability diagram through a charge sensor and noise, will entirely swap the distortions. And the speed of the thresholded algorithm will allow for much larger more diverse training datasets.

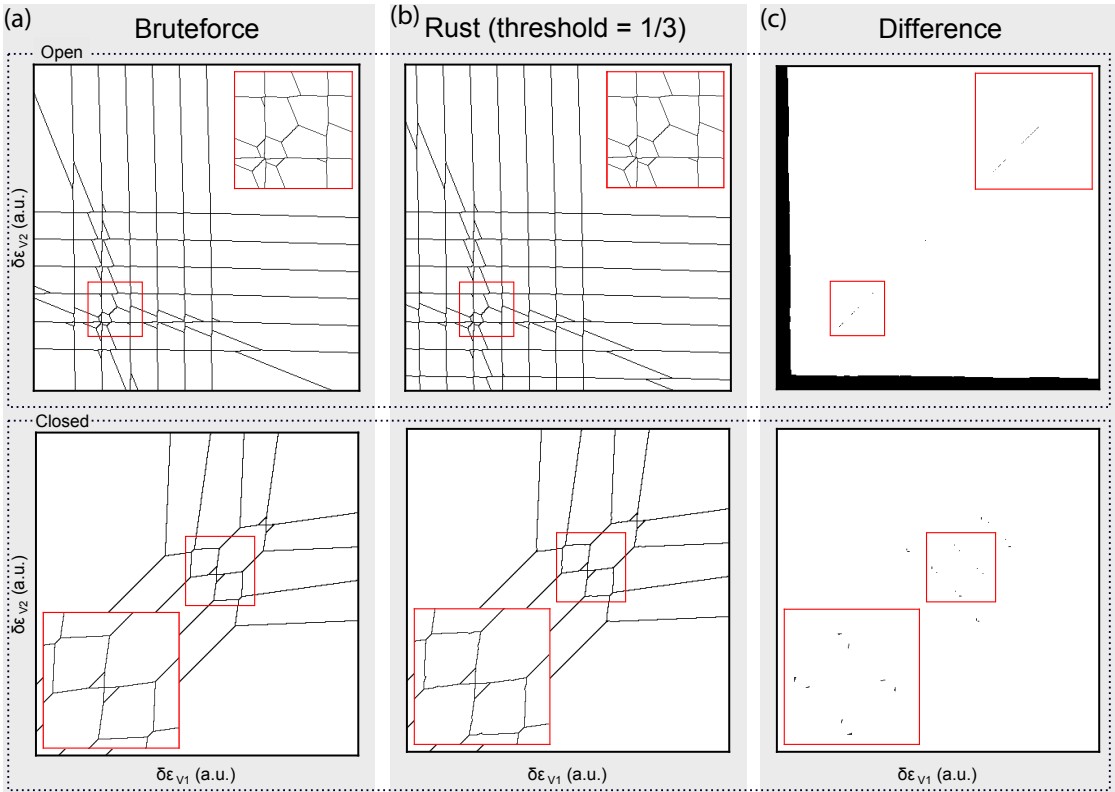

Figure 9. Diagrams illustrating the errors introduced by the thresholding strategy. (a) The charge stability diagram of a four-gate quadrupled quantum dot device as a function of the left and rightmost gates, in both the open and closed regime, where four holes are confined. The brute-force core was used to compute the charge stability diagrams. (b) Identical charge stability diagram, but computed using the Rust core with a threshold of $1/3$. (c) The pixel-wise difference between the brute force and the rust core. Black pixels indicate that the predictions differ. The solid black band along the $x$ and $y$ axis demonstrate the brute-force algorithm failing when $n_{\max}$ is too small. Whilst the small discrepancies in the centre of the plot are due to the thresholding algorithm.

**Appendix K: Simulation times**

This section tabulates the time required to simulate the charge stability diagrams in Fig.5, using each of the software implementations.

| Algorithm | Implementation | Simulation time Figure 5 b) (s) | Simulation time Figure 5 d) (s) |
|---|---|---|---|
| Brute-force | Python | 4.6 | 137.6 |
| Brute-force | JAX | 0.05 + 0.32 jit | 21.1 + 0.4 jit |
| Default | Python | 4.92 | 18.1 |
| Default | JAX | 0.09 + 0.8 jit | 2.1+ 0.8 jit |
| Default | Rust | 0.10 | 0.71 |
| Thresholded $t = 1/2$ | Python | 2.81 | 10.8 |
| Thresholded $t = 1/2$ | Rust | 0.08 | 0.66 |

Table II. The compute times for all the different implementations of the brute-force, default and thresholded algorithms, to recreate plots b) and d) in Figure 5. For the JAX-based implementations, we include the one-off jit compile time. For the brute-force algorithm, we used $n_{max} = 2$ and 5 for the open four-dot array and the closed five-dot array in b) and d), respectively.