# Peer review of "QArray: a GPU-accelerated constant capacitance model simulator for large quantum dot arrays"

_SciPost Physics Codebases, doi:SciPost Phys. Codebases 35-r1.3 (2024) , SciPost Phys. Codebases 35 (2024)_

## Round 1 · Referee Report · Anonymous (Referee 1) · 2024-4-17

Strengths
2 - the proposed method seems to offer a significant advantage over current practice
3 - comparison with experimental data is proposed
4 - the code is carefully commented and accompanied by examples.
Weaknesses
- the code lacks a complete description of all available functionalities and the general structure of the implementation (Doxygen type)
Report
Instead of a brute force search for the ground state involving the study of all possible occupied states, it is proposed to obtain a continuous minimum of the free energy, and then deduce a discrete solution by exploring the neighborhood of this continuous solution, which considerably reduces the algorithmic cost. A second method is also proposed, using a threshold to further reduce the number of configurations to be covered, offering an advantage in some cases.
The stability diagrams obtained experimentally are compared with the method and found to be in agreement. A performance study is also carried out to demonstrate the benefits of the approach. The appendices provide analytical and numerical proofs of the validity of the approximations made by the techniques.
This article is accompanied by a QArray python package whose implementation also uses Rust, a compiled language and offers a GPU implementation with Jax.
Given the editorial quality of the article, the scientific interest associated with it and the package provided, it seems to me that this article following the guidelines of the journal has its place in SciPost Physics Codebases. I would like to suggest a few comments and changes.
Requested changes
1 - Introduction : it would be interesting to have a few sentences in the introduction on the general background to the usefulness of charge stability diagrams.
2 - In Part II, it may be useful to restate the conditions for studying the system in the classical constant capacitance model approach.
3 - It seems to me that the formula for F(q_d,V_g) should also be recalled before injecting equation (3) and obtaining (4).
4 - In part IV .B. equation (10) reintroduce Vg in the arguments of F.
5 - On which configuration is the summation performed to obtain the softmin in the case of thermal enlargement? It seems to me that it refers to all the configurations around the continuous optimum, otherwise the problem back to the brute-force case. This is not very clear in the paragraph.
6 - I think it would be useful to add a few sentences on the benefits of charge sensors, optimal gates voltages and virtual gates. Perhaps a figure would be welcome here.
7 - Part VI: it seems to me that the article is rather allusive on the minimization method used (OSQP). I think this is an important point in the method (when the explicit result cannot be used) and the general principle should be explained. In addition, it doesn't say how precise the continuous optimum must be, how many iterations must be performed, or how much computing time must be devoted to this step.
8 - the origin of the error bars in the figures and the method of obtaining them should be more explicit.
9 - I am uncertain by Figure 5. Smaller values of t give a faster result. However, according to the authors, this is linked to a more approximate result. I think it would be good to be able to see on the figure the competition between calculation time and error committed than just running time.
10 - Appendix C : Reintroduce Q notation, introduce L and lambda
11 - the code lacks a complete description of all available functionalities and the general structure of the implementation (Doxygen type)
12 - I find it confusing that some rust code is in .py and not .rs. In order to see which part of the code is in rust, python or jax, this could typically appear in the documentation
Recommendation
Ask for minor revision

---

## Round 1 · Referee Report · Isidora Araya Day (Referee 2) · 2024-5-14

Strengths
- Implemented useful algorithm for quantum dot array computations
- Faster algorithm than state of art
- Well written and overall clear paper
Weaknesses
- Several modules of the package are not sufficiently tested
- Lacks package documentation
- Unclear if and why the thresholded algorithm is always valid
Report
Summary
The paper introduces a Python package and two useful algorithms for finding the ground state configuration of weakly coupled quantum dots. This is a costly computation relevant to the quantum dot arrays community. This work reduces the time costs of such simulations significantly, allowing for the creation of larger data sets and data analysis.
Because the paper introduces a software package, QArray, I have additionally reviewed its code. The Github repository describes the package, indicates how to install it, and contains the package's license, automated tests, examples, and entire code.
The package has unit tests that are run systematically using Github actions. I have, however, found that several files have large untested amounts of code. This can be confirmed running "pytest --cov --cov-report=html". While some of these files contain only a few functions, some of these are important functions for the package functionality, computing ground states and charge configurations. Without tests, the package's code may become unreliable over time and its current functionality is hard to validate.
While most of the code in the repository has comments about usage and input types, several public functions lack docstrings. This can be confirmed installing ruff and running "ruff check . --select D103". Documentation may increase the package's visibility and help potential users and contributors. The examples currently in the repo contain code, but lack context to serve as documentation. Having complete documentation is a requirement for the acceptance criteria of SciPost Physics Codebases.
Questions
- Is the integer minimum unique? If the configurations of two dots is completely symmetric with respect to gate voltages, would the default algorithm and thresholded algorithm find both configurations or would they only find one? It is not clear to me that the thresholded algorithm always returns the correct integer number configuration.
- Is there an intuitive way to understand why the thresholded algorithm works for any potential? Fig 2b delivers the point, but the potential is fairly specific.
- Is it possible to combine quantum dots where some are electron dots and others are hole dots? The fact that $\vec{Q} = \pm e \vec{N}$ suggests not, because elements $\vec{N}$ are defined as integers. Is this a limitation?
Requested changes
- I would like to ask the authors to improve and extend the package tests.
- I would like to ask the authors to document public functions and write tutorials, not just examples. The authors could also consider writing package documentation, e.g. using https://readthedocs.org/.
- Not all figures are referenced in the main text, e.g Fig. 1a and 1b., and Section II would benefit from referring to Fig. 1a. Please ensure that all figures are referenced.
- Gray and black lines in Fig. 1a are hard to distinguish. I would like to suggest using a different color or dashing one of the lines.
- The paper contains a few typos ("Compted"), please fix these.
- The references are not formatted, please format them.
Recommendation
Ask for major revision

---

## Round 1 · Referee Report · Anonymous (Referee 3) · 2024-5-30

Strengths
- new algorithm with improved runtime
- python interface with optimized backends allowing for parallelization on CPU and GPU
- unit tests
Weaknesses
- documentation of functions often lacking
- validity of the improved algorithms sometimes unclear (see report)
- some parts of manuscript/usage of the package are hard to follow (see report)
Report
The paper describes a package to efficiently compute the ground state of quantum dot arrays within the approximation of the constant capacitor model. In addition, it contains functionality to simulate the response of an actual experimental probe, such as a charge sensing dot.
This is a topic of great current interest (two similar packages were published almost simultaneously), driven by the interest to generate artificial training data for machine learning. The algorithm shows a clear improvement over the standard brute-force method, and the implementation has an emphasis on efficiency, including options to parallelize on a CPU and GPU. The package overall seems well-organized, with a structured API and useful comments in the code.
I installed the package on my personal laptop (Ubuntu 22.04) using conda. The command 'pip install qarray' worked after I also installed 'cmake' via 'conda install cmake'. (See some comments in requested changes with regards to installation). I then ran the example provided in the main text (copied from the README on the github page), and tried another of the examples provided on github. These ran without any problems.
In general, I believe that in principle this package and accompanying paper should be publised in SciPost Physics Codebases. However, currently the paper and code are lacking with regards to the following acceptance criteria:
- At least one example application must be presented in detail
- The documentation must be complete, including detailed instructions on downloading, installing and running the software
In particular, I encountered the following problems: - Many functions in the code are not documented properly. Moreover, many docstrings are very short and hard to understand. Take the example of the function 'charge_state_contrast'. The docstring only says 'Function to compute the charge state contrast'. It is missing an explanation though what charge state contrast even is (google only finds 5 instances for this expression, so it is not widely used). I can guess it from the result, but I should not need to guess. - I don't find the package very intuitive to use. I can run the examples, but I would find it hard to code my own simulation. A reason for this is that the example in the paper is very short and incomplete (not even the output is shown). The different concepts used by the classes that build the system are barely explained. There are many examples in the repository, but these also lack a proper tutorial-like documentation that would explain how to build the system, and how to use the different functions. - Moreover, for the advanced examples shown in Fig. 3, I found it hard to find the corresponding code in the repository: I eventually found the code for Fig. 3b by clicking on names that seemed relevant, but I gave up to find Fig. 3a after 5 minutes. It should not be that hard to find these examples. - The package provides functionality to mimic experimental signatures, but I find this intransparent: The description in IV.A is very hard to follow. What is exactly done? Can this be explained with formulas/sketches? Moreover, Fig. 3b does some tricks to simulate the response to a QPC, but this is only explained in a single sentence in the caption - The authors argue for the validity of their algorithm in the appendix, and show limits on the condition number of the capacitance matrix. They argue that this is usually fulfilled in experiment, but does the code actually check this criterion? Given that correctness is essential for the package, the default algorithm must be completely robust, and this check seems essential. - I find it hard to judge the threshold algorithm. Is it possible to warn the user if the threshold is wrong, or is it up to the user completely? There are estimates for thresholds, but they are not good enough to be used automatically?
Requested changes
- Improve docstrings in code
- Explain usage of code better (either in paper or in a separate documentation): add tutorials, explain the concepts to build the system better, show several explicit examples with extended explanation
- Make code of all examples in the paper easy to find
- Explain in detail how the experimental signatures (charge sensor dot) are calculated. Note that I found the term ''non-charge sensor dot'' very confusing. I guess you mean the regular dots?
- Eq. C.3 talkes about differentiating L, but there is no L introduced in C.
- address the issues regarding validity mentioned in the report.
Recommendation
Ask for major revision

---

## Round 2 · Author Response

In response to questions from review 2
Question 1:Is the integer minimum unique? If the configurations of two dots is completely symmetric with respect to gate voltages, would the default algorithm and thresholded algorithm find both configurations or would they only find one?
Response: We have now clarified this point in the manuscript. Along a charge transition (shown as black lines in Figures 1 c and d), two charge states are indeed equal in energy. At T = 0 this constitutes an infinitesimally thin line. At these points, our algorithm would find only one of the change states due to the limits of numerical precision. However, at finite temperatures, the return charge state would be an equal mixture of the two.
Question 2:It is not clear to me that the thresholded algorithm always returns the correct integer number configuration. Is there an intuitive way to understand why the thresholded algorithm works for any potential? Fig 2b delivers the point, but the potential is fairly specific.
Response:We agree with the reviewer that, indeed, it is not true that the thresholded algorithm always returns the correct charge state(see Figures 7c, 8 and 9). As we discussed in the appendix, we observe artefacts particularly as distortions of the interdot charge transitions. Importantly, our results show that a correct choice of threshold can often reduce the simulation time without introducing errors. Based on comments from other reviewers, we have added a warning that advises the user to either increase the threshold or use the default algorithm if the value selected is found to be too low, based on eq. (F3). To further clarify this aspect, we have also added a new plot showing an example of error against the threshold value.
Regarding the reviewer’s second question, about the validity of the threshold algorithm for any potential, we claim that the class of potential such as those of figure 2b are indeed quite general, given the assumptions of the constant capacitance model. About this, we note that the prediction of the capacitance model is generally valid for weakly coupled quantum dots. For example, it is well known that double quantum dots that are interacting via a strong tunnel coupling exhibit clear deviations from the model predictions at the interdot charge transitions see (Ref. 1). For weak interdot coupling, assuming that the capacitive coupling between dots is weaker than the coupling between the dots and the gates, is also a valid assumption. As a result, the matrix $c_{dd}$ (and therefore $c^{−1}_{dd}$) must be approximately diagonal, leading to a fairly general class of potentials such as that of Fig 2b.
Question 3: Is it possible to combine quantum dots where some are electron dots and others are hole dots? The fact that Q⃗ = ±eN⃗ suggests not because elements N⃗ are defined as integers. Is this a limitation?
Response: The ability to combine both electrons and holes within the same array is not currently available in our software. This feature could be implemented in future versions of the software. We will work on this.
In response to questions from review 3:
Comment 1: Many functions in the code are not documented properly. Moreover, many docstrings are very short and hard to understand. Take the example of the function ”charge state contrast”. The docstring only says ”Function to compute the charge state contrast”. It is missing an explanation though what charge state contrast even is (google only finds 5 instances for this expression, so it is not widely used). I can guess it from the result, but I should not need to guess.
Response: We thank the reviewer for the valuable feedback. We have improved the doc-strings and documentation for the project and included a short description of each function’s functionality.
Comment 2: I don’t find the package very intuitive to use. I can run the examples, but I would find it hard to code my own simulation. A reason for this is that the example in the paper is very short and incomplete (not even the output is shown). The different concepts used by the classes that build the system are barely explained. There are many examples in the repository, but these also lack a proper tutorial-like documentation that would explain how to build the system, and how to use the different functions.
Response: We agree with the reviewer on the usefulness of a more tutorial like documentation and we apologise for not including this earlier. We have now added a ”read the docs” style documentation page (https://qarray.readthedocs.io/en/latest/index.html). This set of step-by-step explanations contains several examples that we hope will be useful to get new users ready to start coding their own models.
Comment 3: Moreover, for the advanced examples shown in Fig. 3, I found it hard to find the corresponding code in the repository: I eventually found the code for Fig. 3b by clicking on names that seemed relevant, but I gave up to find Fig. 3a after 5 minutes. It should not be that hard to find these examples.
Response: We agree with the reviewer and we apologise for this oversight. We have now renamed the code to reproduce these figures to fig_3b.py and fig_3d.py, respectively.
Comment 4: The package provides functionality to mimic experimental signatures, but I find this intransparent: The description in IV.A is very hard to follow. What is exactly done? Can this be explained with formulas/sketches? Moreover, Fig. 3b does some tricks to simulate the response to a QPC, but this is only explained in a single sentence in the caption
Response: We thank the reviewer for this comment. We have now clarified this in section IV.A and improved the description of the charge sensing simulation.
Comment 5: The authors argue for the validity of their algorithm in the appendix and show limits on the condition number of the capacitance matrix. They argue that this is usually fulfilled in experiments, but does the code actually check this criterion? Given that correctness is essential for the package, the default algorithm must be completely robust, and this check seems essential.
Response: We thank the reviewer for rising this important point. We have now added the suggested check in the revised version of the code.
Comment 6: I find it hard to judge the threshold algorithm. Is it possible to warn the user if the threshold is wrong, or is it up to the user completely? There are estimates for thresholds, but they are not good enough to be used automatically?
Response: We see the concern of the reviewer that the approximation made by the threshold algorithm may cause artefacts in the simulation. Following the suggestions also from other reviewers, in the revised version of our manuscript and code we have included a new figure and a warning to advise the user of the potential discrepancies and artefact that may arise using the thresholded algorithm. The new figure shows the discrepancy between a simulation with and without the threshold, as a function of the threshold value. As already mentioned in other reviewer’s comments, the newly introduced warning notifies the user when the selected threshold is below a value estimated from the capacitance matrix following Eq. (F3).

---

## Round 2 · List of Changes

Changes to the manuscript are written in blue in the updated manuscript version and marked with “>[…]<“ here.
In answer to review 1:
Requested change 1: Introduction : it would be interesting to have a few sentences in the introduction on the general background to the usefulness of charge stability diagrams.
Response: We thank the reviewer for this comment. As requested, we have modified the first paragraph of the manuscript.
Change in ‘I. INTRODUCTION’:
“Semiconductor quantum dot arrays are one of the leading platforms for the realisation of large-scale quantum circuits. As they grow in size and complexity, these arrays present exciting opportunities and significant challenges. >Tuning and operating these devices involves finding precise gate voltages to achieve the desired charge occupancy and interdot coupling parameters. Charge stability diagrams provide a visual representation of the charge occupation in the quantum dot as a function of the applied gate voltages, allowing the identification and control of transitions between different charge configurations.<
The constant capacitance model is a widely-used equivalent circuit framework that models the electrostatic characteristics of quantum dot arrays. >This model can be used to compute charge stability diagrams of quantum dots arrays [1-7]. As a result, the constant capacitance model< has proven to be an invaluable resource for gaining insights into complex charge stability diagrams and training neural networks for automated tuning strategies [2,5,8-11]. ”
Requested change 2: In Part II, it may be useful to restate the conditions for studying the system in the classical constant capacitance model approach.
Response: We have included the necessary condition for the constant capacitance model as per ref: [Rev. Mod. Phys. 79, 1217 (2007)] .
Change in “II. THE CONSTANT CAPACITANCE MODEL”:
“ The constant capacitance model describes an array of quantum dots and their associated electrostatic gates as nodes in a network of fixed capacitors [1–7], >(see Fig. 1a). The model is based on the assumption that a single constant capacitance C models the interactions between the charges on a single quantum dot and those in the environment and that the single-particle energy levels do not depend on the number of charges considered. Additionally, the model is a good approximation when the quantum dots are weakly coupled enough that quantum tunnelling effects can be ignored. Typically, the constant capacitance model is valid in a small range of gate voltages, as significant changes in the confinement potential can shift the dot positions and consequently alter their capacitive couplings. <”
Requested change 3: It seems to me that the formula for F(qd,Vg) should also be recalled before injecting equation (3) and obtaining (4).
Response: We thank the reviewer for this suggestion; we have updated the paragraph.
Change in “II. THE CONSTANT CAPACITANCE MODEL”:
“The free energy of the system, > $F(\vec{Q}; \vec{V}) = U(\vec{Q}; \vec{V}) - W(\vec{Q}; \vec{V}) = \vec{V}^T \vec{Q} / 2 - \vec{Q}_g \vec{V}_g$ < , can then be written as
\begin{equation}\tag{4}
F(\vec{Q}_d; \vec{V}_g) = \frac{1}{2}\vec{Q}_d^{T} c_{dd}^{-1} Q_d - (c_{dd}^{-1} c_{cd} \vec{V}_g)^{T} \vec{Q}_d,
\end{equation}”
Requested change 4: In part IV .B. equation (10) reintroduce Vg in the arguments of F.
Response: As requested, we have added the arguments to the function. Please note this is now equation (11) in the revised manuscript.
Change in “IV. ADDITIONAL FUNCTIONALITY”:
We have added the arguments to the function, which is now equation (11).
Requested change 5: On which configuration is the summation performed to obtain the soft- min in the case of thermal enlargement? It seems to me that it refers to all the configurations around the continuous optimum, otherwise the problem back to the brute-force case. This is not very clear in the paragraph.
Response: We agree with the reviewer, and we have now clarified this in the manuscript.
Change in “IV. ADDITIONAL FUNCTIONALITY”:
“The summation runs over all the allowed charge configurations >considered by the algorithm<. For a closed array, this means eliminating the charge states with the total number of charges different than $\hat{N}$. >This approximation is valid for temperatures such that $k_B T$ is smaller than the dot charging energies, since the contribution of other charge states is exponentially suppressed.<”
Requested change 6: I think it would be useful to add a few sentences on the benefits of charge sensors, optimal gates voltages and virtual gates. Perhaps a figure would be welcome here.
Response: We agree with the reviewer that highlighting the benefits of these functionalities will provide more context to the paper, and we have added a short paragraph.
Change in “IV. ADDITIONAL FUNCTIONALITY”:
“>These functionalities will be useful both when using QArray as a tool to understand charge stability diagrams and for automated tuning strategies. The charge sensing functions can be used to simulate realistic charge stability diagrams measured with a charge sensor. The optimal gate voltage function can be used to simulate a voltage sweep centred around a chosen charge state for any given capacitance matrix. Given a specific capacitance matrix, the optimal virtual gate function can be used to find the linear combination of gates that can modify the electrochemical potential of a single dot compensating for the effects of voltage cross-talk.<“
Requested change 7: Part VI: it seems to me that the article is rather allusive on the minimization method used (OSQP). I think this is an important point in the method (when the explicit result cannot be used) and the general principle should be explained. In addition, it doesn’t say how precise the continuous optimum must be, how many iterations must be performed, or how much computing time must be devoted to this step.
Response: We thank the reviewer for this insightful comment. We used OSQP since we found it was the highest-performance solver for the specific class of quadratic-constrained optimisation problems addressed in our manuscript. In the main manuscript, we have now included a new sentence that clarifies the choice of the solver and directs the interested reader to a more specialised reference (written by the solver’s authors).
Further on the reviewer comment: The number of performed iterations is determined by the solver such that the relative and absolute error in the solution falls below $10^{−3}$ (the default solver value). Given that our algorithms iterate over the nearest integer charge states, we find this precision acceptable.
Regarding the computational time, our benchmarks found the OSQP solver able to identify the continuous minimum of double, triple and even quadruple dot systems in less than 10 μs.
We have now clarified these aspects in the dedicated section of the manuscript (Section 3.A.2).
Change in “VI. IMPLEMENTATIONS”:
“>For a complete description of the OSQP solver and its benchmarks, see Ref. [27]. For the solver’s hyperparameters, we use the default values suggested by this reference. As a result, the solver performs the required number of iterations to obtain the continuous minimum to within an absolute and relative error of $10^{−3}$. The solver updates the step size automatically. Our benchmarks found that the OSQP solver was able to find the continuous minimum of double, triple and even quadruple dot systems in less than 10 μs per voltage configuration.<“
Requested change 8: the origin of the error bars in the figures and the method of obtaining them should be more explicit.
Response: We thank the reviewer for pointing this and we apologise for the oversight. We have now added a short description in the figure caption.
Change in “VII. BENCHMARKS”, in the caption of Figure 6:
“>The error bars represent confidence intervals of 2σ, such that for a randomly chosen set of capacitance matrices and voltages, the compute time should fall within the error bars 95% of the time. <”
Requested change 9: I am uncertain by Figure 5. Smaller values of t give a faster result. However, according to the authors, this is linked to a more approximate result. I think it would be good to be able to see on the figure the competition between calculation time and error committed than just running time.
Response: We agree with the reviewer that this would be rather informative and we have thus added the requested figure to the main manuscript (see new Fig. 7(c)). In addition, we wish to note that the thresholding strategy errors are further discussed in the appendix section J, see also Fig. 8 and 9. As specified in the paper, we noted that the threshold value after which significant discrepancies appear in the charge stability diagram depends on the parameters of the capacitance matrix. Hence, as further discussed in the appendix, a suitable choice of the threshold should be made by taking these parameters into account. To further clarify these points to the reader, we have added a new panel to (now) figure 7 of the main manuscript showing the percentage error introduced by the thresholded algorithm as a function of the chosen threshold. We estimated this error as a pixel comparison against the default (non-thresholded) version.
Finally, following also the advice of reviewer 3, we have coded explicit warnings for the user when the threshold is set too low based on equation (F3) of the appendix.
Requested change 10: Appendix C : Reintroduce Q notation, introduce L and lambda
Response: We have introduced the Q notation as requested. The references to L and λ were not meant to be there, so these have been removed.
Change in “Appendix C”:
New introduction to Q notation. The references to L and λ have been removed.
Requested change 11: the code lacks a complete description of all available functionalities and the general structure of the implementation (Doxygen type)
Response: Based on the feedback from all reviewers, we have substantially improved the documentation. There is an API overview of the whole package as part of this documentation.
Requested change 12: I find it confusing that some rust code is in .py and not .rs. In order to see which part of the code is in rust, python or jax, this could typically appear in the documentation.
Response: We assume the reviewer is referring to some thin Python wrapper functions present in the Qarray rust implementation folder. If so, we apologise for the confusion.
To clarify this point, all rust code is packaged as a separate Python package called “qarray- rust-core” (https://pypi.org/project/qarray-rust-core/). This package is a dependency of the QArray package. All code in this package is rust code (with .rs file names). The package is compiled using maturin (https://www.maturin.rs) so that it can be called by Python. The wrapper functions responsible for this confusion merely perform type conversions, such as f32 to f64, in the numpy arrays to be passed.
In answer to review 2:
Requested change 1:I would like to ask the authors to improve and extend the package tests.
Response: As requested by the reviewer, we have now expanded the testing suite such that 100% of the files and over 90% of code lines are covered by at least one test.
Requested change 2:I would like to ask the authors to document public functions and write tutorials, not just examples. The authors could also consider writing package documentation, e.g. using https: // readthedocs. org/ .
Response: Following the reviewer suggestion we have now written the readthedocs documentation, available at https://qarray.readthedocs.io/en/latest/introduction.html
Requested change 3:Not all figures are referenced in the main text, e.g Fig. 1a and 1b., and Section II would benefit from referring to Fig. 1a. Please ensure that all figures are referenced.
Response: We thank the reviewer for spotting this and we apologise for this oversight, we have now referenced all the figures in the main manuscript.
“Within QArray, the brute-force and our default algorithm are implemented in Python, Rust and JAX, >(see Fig. 1b)<.”
“The constant capacitance model describes an array of quantum dots and their associated electrostatic gates as nodes in a network of fixed capacitors [1–7], >see (Fig. 1a)<”.
Requested change 4: Gray and black lines in Fig. 1a are hard to distinguish. I would like to suggest using a different color or dashing one of the lines.
Response:We have modified the figure as requested.
Requested change 5: The paper contains a few typos (”Compted”), please fix these.
Response: We thank the reviewer for pointing this to us, we have fixed these typos in our revised manuscript.
Requested change 6: The references are not formatted, please format them.
Response: We have now reformatted the references.
In answer to review 3:
In answering comment 4, we have changed the following in IV.A:
“ To simulate charge-sensing measurements, we >distinguish between charge-sensing dots and regular dots. We first compute the charge state that minimises the free energy of the regular dots. With the charges on the regular dots fixed, we then determine the charge configuration on the sensor dots that minimises the overall free energy.
Next, we evaluate the difference in free energy between the minimum free energy charge state of the regular dots and the sensor's charge state next lowest in energy, which we denote as \(\Delta F(\vec{V}_g)\). We assume that the charge sensor dots are in the strongly coupled regime, so the conductance follows the Breit-Wigner formula [21]. Consequently, the charge sensor's response is Lorentzian, and we model the sensor response as:
\begin{equation}
s(\vec{V}_g) = \frac{1}{(\Delta F(\vec{V}_g) / \Gamma) ^2 + 1}
\end{equation}
where \(\Gamma\) is the sum of the tunnel rates to the source and drain; this parameter is configurable by the user to match their experimental setup. Additionally, we include various noise models to simulate white noise, \(1/f\) noise, and sensor switches due to fluctuating charge traps, among others.< ”
Requested change 1: Improve docstrings in code
Response: We have now updated the docstring to include descriptions of the functions, all their arguments and return values.
Requested change 2: Explain usage of code better (either in paper or in a separate documentation): add tutorials, explain the concepts to build the system better, show several explicit examples with extended explanation
Response: We have rewritten the example section to make it more user-friendly. In addition, on the readthedocs documentation page, we have written a set of examples covering:
1. Getting started
2. Using the GUI
3. Simulating change sensing measurements
4. Adding noise to the change sensing measurements
5. Adding latching to the change sensing measurements.
Requested change 3: Make code of all examples in the paper easy to find
Response: We thank the reviewer for highlighting this and we apologise for the oversight. The code examples are now clearly named figure 4b and figure 4d and are in the examples folder. https://github.com/b-vanstraaten/qarray/tree/main/examples
Requested change 4: Explain in detail how the experimental signatures (charge sensor dot) are calculated. Note that I found the term ”non-charge sensor dot” very confusing. I guess you mean the regular dots?
Response: We have made the requested change. Please see our reply on the reviewer comment 4.
Requested change 5: Eq. C.3 talks about differentiating L, but there is no L introduced in C.
Response: We thank the reviewer for noticing this, this was a typo. We apologise for the oversight and we have now fixed it.
Requested change 6: address the issues regarding validity mentioned in the report.
Response: We have made the requested changes, as per our answer to comment 5 of the reviewer we have now added these checks to the code.

---

## Editorial Decision

published